# Serine Hydroxymethyltransferase Modulates Midgut Physiology in *Aedes aegypti* Through miRNA Regulation: Insights from Small RNA Sequencing and Gene Expression Analysis

**DOI:** 10.3390/biom15050644

**Published:** 2025-04-30

**Authors:** Qian Pu, Yujiao Han, Zhuanzhuan Su, Houming Ren, Qingshan Ou, Symphony Kashyap, Shiping Liu

**Affiliations:** State Key Laboratory of Resource Insects, Southwest University, Chongqing 400715, China; pq7426@email.swu.edu.cn (Q.P.); hyj1998@email.swu.edu.cn (Y.H.); su2002@email.swu.edu.cn (Z.S.); rhm7833097@email.swu.edu.cn (H.R.); oqs123033@email.swu.edu.cn (Q.O.); symphonykashyaprrl@gmail.com (S.K.)

**Keywords:** *Aedes aegypti* mosquitoes, serine hydroxymethyltransferase (SHMT), microRNA, midgut, blood meal

## Abstract

*Aedes aegypti* mosquitoes are critical vectors of arboviruses, responsible for transmitting pathogens that pose significant public health challenges. Serine hydroxymethyltransferase (SHMT), a key enzyme in one-carbon metabolism, plays a vital role in various biological processes, including DNA synthesis, energy metabolism, and cell proliferation. Although *SHMT* is expressed at low levels in the midgut of *Aedes aegypti*, its silencing has been shown to inhibit blood meal digestion. The precise mechanisms by which *SHMT* regulates midgut physiology in mosquitoes remain poorly understood. In this study, we employed small RNA sequencing and quantitative PCR to identify differentially expressed miRNAs (DEMs) following *SHMT* downregulation. We focused on a subset of DEMs—miR-2940-5p, miR-2940-3p, miR-2941, and miR-306-5p—to explore their potential biological functions. To further elucidate the molecular mechanisms underlying the miRNA response to *SHMT* downregulation, we analyzed the expression levels of key genes involved in the miRNA biogenesis pathway. Our results demonstrated that several critical enzymes, including Drosha, Dicer1, and AGO1, exhibited significant changes in expression upon *SHMT* silencing. This study provides new insights into the molecular mechanisms through which *SHMT* influences the biological functions and nutritional metabolism of the mosquito midgut. By linking *SHMT* activity to miRNA regulation, our findings highlight a potential pathway by which *SHMT* modulates midgut physiology, offering a foundation for future research into mosquito biology and vector control strategies.

## 1. Introduction

The *Aedes aegypti* (*Ae. aegypti*) mosquito is a primary vector for several arboviruses, including dengue, Zika, chikungunya, and yellow fever, which collectively pose significant threats to global public health. These pathogens are responsible for millions of infections annually, with dengue alone causing an estimated 390 million infections each year [1,2]. Female *Ae. aegypti* mosquitoes require blood meals to produce eggs, making their feeding behavior a critical factor in disease transmission [3]. The rapid reproduction rate and adaptability of *Ae. aegypti* enable swift population recovery following control interventions, perpetuating the cycle of disease transmission [4,5,6]. This resilience is further compounded by the mosquito’s ability to develop resistance to commonly used insecticides, such as pyrethroids and organophosphates, which complicates eradication efforts [7,8]. The mosquito midgut plays a central role in blood meal digestion and nutrient absorption, processes that are essential for egg development and overall mosquito fitness. The midgut is not only a site for nutrient processing but also a critical barrier against pathogen invasion [9,10]. After a blood meal, the midgut undergoes significant physiological changes, including the upregulation of digestive enzymes such as trypsins and aminopeptidases, which break down proteins into amino acids for absorption [11,12]. These nutrients are then utilized for vitellogenesis, the process of yolk protein production in the ovaries, which is essential for egg maturation. Given that mosquito life activities are meticulously regulated at the genetic level, elucidating the molecular mechanisms governing these processes is crucial for developing effective population control measures.

MicroRNAs (miRNAs) are small non-coding RNAs that play pivotal roles in post-transcriptional gene regulation. By binding to target mRNAs, miRNAs can modulate gene expression, thereby influencing a wide range of biological processes, including development, metabolism, and immune responses [13,14,15]. A substantial body of evidence indicates that miRNAs are essential for regulating the reproductive and nutritional cycles of mosquitoes, thereby influencing their growth and development [16], blood feeding and digestion [17,18], ovarian maturation [19,20], immune responses [21,22], and other critical physiological processes [23,24]. In our previous study, we identified that mosquito- and midgut-specific miR-1174 (MIMAT0014265) plays a crucial role in modulating the biological functions of the mosquito midgut through its targeted regulation of serine hydroxymethyltransferase (*SHMT*; AAEL002510) [25]. Comprehensive proteomics and metabolomics analyses have revealed that miR-1174 is a pivotal regulator across multiple metabolic pathways, encompassing amino acid metabolism, nucleotide metabolism, fatty acid metabolism, and carbohydrate metabolism; consequently, the aberrant phenotypic manifestations observed in mosquitoes lacking miR-1174 are likely attributable to the cumulative impact of alterations within these metabolic pathways [26]. These findings underscore the significant biological relevance of the regulatory axis established between miR-1174 and its target gene *SHMT*, particularly in processes critical to mosquito survival and reproduction, such as blood feeding, digestion and absorption of nutrients, metabolic regulation, and reproductive activities.

SHMT is an evolutionarily conserved enzyme that plays a crucial role in cellular metabolism by catalyzing the reversible interconversion of L-serine and tetrahydrofolate to glycine and 5,10-methylenetetrahydrofolate [27,28]. This enzyme is pivotal to various biological processes, including development, immune function, and oncogenesis [28,29,30,31,32,33]. *SHMT* is expressed at low levels in the midgut but plays a crucial regulatory role in the digestion of blood. Specifically, digestive enzymes that are highly expressed in a tissue-specific manner within the midgut respond to the *SHMT* downregulation at the transcriptional level, indicating its regulatory effect on the expression of these enzymes [34]. The spatial expression separation of *SHMT* and digestive enzymes suggests that *SHMT* likely indirectly influences the biological functions of the mosquito midgut. Therefore, there may be intermediate signaling pathways or regulatory molecules between SHMT and digestive enzymes. Given the diverse regulatory roles of miRNAs, it is possible that *SHMT*-responsive miRNAs are involved in modulating the physiological functions of the midgut.

Through small RNA sequencing and quantitative PCR analyses, we identified differentially expressed miRNAs (DEMs) in response to *SHMT* downregulation. Furthermore, we conducted preliminary functional investigations of selected DEMs to explore their potential biological roles. To elucidate the molecular mechanisms underlying miRNA response to *SHMT* modulation, we performed systematic evaluation of key components in the miRNA biogenesis pathway. Our results demonstrated that several critical enzymes involved in miRNA synthesis, including Drosha (AAEL008592), Dicer1(AAEL001612), and AGO1 (AAEL012410), exhibited significant responses to *SHMT* downregulation. This study provides novel insights into the molecular mechanisms through which *SHMT* regulates both the biological functions and nutritional metabolism of the mosquito midgut, establishing a foundation for further exploration of the complex regulatory networks involving *SHMT* and miRNAs in mosquito physiology.

## 2. Materials and Methods

### 2.1. Mosquito Rearing and Sample Collection

*Aedes aegypti* (NIH Rockefeller strain) mosquitoes utilized in this study were bred as previously described [26]. Specifically, the *Ae. aegypti* colony was maintained at 28 °C with 75% relative humidity under a 12-h light/dark photoperiod. Larvae were reared in water supplemented with an artificial diet consisting of a 1:1 mixture of yeast and rat chow. Adult mosquitoes were provided with cotton pads soaked in water and a 10% glucose solution both before and after blood feeding. Three to four-day-old female mosquitoes were blood-fed on white rats that had been anesthetized with isoflurane. Laboratory vertebrate animals were handled in accordance with guidelines approved by the Southwest University Animal Care and Use Committee.

To assess the blood feeding rate, three independent cages were established for both control and treatment groups, representing three biological replicates. Each cage contained a population of 40 female mosquitoes accompanied by a minimum of 20 male mosquitoes. Following the blood feeding period, the number of engorged females in each cage was recorded. The blood feeding rate for each replicate was calculated using the following equation: Blood feeding rate (%) = (Number of blood-fed females/Total number of females) × 100. Statistical analysis was subsequently performed using GraphPad Prism 8 (GraphPad Software, San Diego, CA, USA).

To collect mosquito individuals and tissues post-blood meal, the blood bolus was excised from the gut using forceps under a stereomicroscope. Dissections of mosquito tissues were conducted in phosphate-buffered saline (PBS). All experiments in this study were exclusively performed on adult female mosquitoes.

### 2.2. DsRNA Synthesis Analysis

Total RNA was extracted from mosquitoes by TRIzol Reagent (Invitrogen, CA, USA). Reverse transcription was performed with PrimeScript^TM^ RT reagent Kit with gDNA Eraser (Takara Bio, Shiga, Japan). DNA templates for dsRNA synthesis were synthesized using PrimerSTAR enzyme (Takara Bio, Shiga, Japan), and the sequence accuracy was confirmed through sequencing analysis. The dsRNA of *SHMT* (ds*SHMT*) and *EGFP* (ds*EGFP*) were synthesized using MEGAscript RNAi Kit (Ambion, Austin, TX, USA) following the manufacturer’s manual. A total of 1500–2000 ng/μL of dsRNA solubilized in 0.5 μL of nuclease-free water was injected into each female adult mosquito at 15–20 h post-eclosion (PE), as described previously [34]. All primers for dsRNA synthesis are listed in the Appendix A.

### 2.3. Real-Time Fluorescence Quantitative PCR (RT-qPCR)

Total RNA was extracted from samples using TRIzol Reagent (Invitrogen, Carlsbad, CA, USA) according to the manufacturer’s protocol and subsequently quantified using a Nano-300 Spectrophotometer (Thermo Fisher Scientific, Waltham, MA, USA). The integrity of the isolated RNA was assessed via 1% agarose gel electrophoresis. For the analysis of protein-coding gene expression, 1 µg of total RNA was reverse-transcribed into complementary DNA (cDNA) using the PrimeScript^TM^ RT reagent Kit with gDNA Eraser (TaKaRa Bio Inc., Otsu, Shiga, Japan). Quantitative real-time PCR (RT-qPCR) was performed using NovoStart^®^ SYBR qPCR SuperMix Plus (Novoprotein, Suzhou, Jiangsu, China) on an Applied Biosystems 7500 Fast Real-Time PCR System (Applied Biosystems, Thermo Fisher Scientific, Waltham, MA, USA), with ribosomal protein S7 (RPS7; AAEL009496-RA) serving as the internal control. The thermal cycling conditions were as follows: initial denaturation at 95 °C for 1 min, followed by 40 cycles of 95 °C for 20 s and 60 °C for 1 min, concluding with a dissociation curve analysis at 95 °C for 15 s, 60 °C for 1 min, and 95 °C for 15 s.

To examine the expression level of miRNAs, 1 μg of total RNA was reverse-transcribed into cDNA using the Hifair^®^ miRNA 1st Strand cDNA Synthesis Kit (YESEN, Shanghai, China). Quantitative real-time PCR (RT-qPCR) was performed using a Hieff^®^ miRNA UniversalqPCR SYBR Master Mix (YESEN, Shanghai, China). DEMs were quantified using TB Green Premix Ex Taq^TM^ II (Takara Bio, Shiga, Japan) on an Applied Biosystems 7500 Fast Real-Time PCR System (Applied Biosystems, Thermo Fisher Scientific, Waltham, MA, USA). The amplification protocol consisted of an initial denaturation at 95 °C for 30 s, followed by 40 cycles of denaturation at 95 °C for 5 s and annealing/extension at 60 °C for 34 s, concluding with a melt curve analysis comprising steps at 95 °C for 15 s, 60 °C for 1 min, and a final step at 95 °C for 15 s. Small nuclear RNA *U6* (AAEL029000-RA) was utilized as the internal control. All primers used for RT-qPCR are listed in the Appendix A. Statistical analysis of RT-PCR data was conducted in accordance with Methods Section 2.7.

### 2.4. Sample Preparation for Small RNA Sequencing

About 0.5 µL of ds*SHMT* (2000 ng/µL) was injected into each female mosquito at about 20 h PE. Under the cultivation conditions of 27–28 °C, the injection wounds of mosquitoes were completely healed approximately 3 days post-injection (72 h PI), and the mosquitoes resumed normal physiological activities, including blood feeding capacity. Therefore, we consider this time point suitable for sample collection to avoid interference from injection trauma on experimental outcomes. Prior to blood feeding, mosquitoes with SHMT knockdown did not exhibit any obvious abnormal phenotypes. However, approximately 15 h post-blood meal (15 h PBM), phenotypes such as delayed blood digestion and ovarian development began to emerge, indicating that this time point is appropriate for detecting early molecular changes following a blood meal. Consequently, this study collected samples at 72 h PI (before blood meal) and 15 h PBM for sequencing and quantitative validation. Small RNA sequencing was performed at Novo Biogenic on the Illumina 6000 novoseqse50 platform.

### 2.5. Small RNA Sequencing Analysis

#### 2.5.1. Quality Control of Sequencing Data

The raw sequencing data underwent initial quality control analysis. The mismatch rate was below 0.01%, the Q20 scores exceeded 98%, the Q30 scores were consistently above 95%, and the GC content was approximately 48% (Appendix A). To ensure high-quality clean reads, sequences containing more than 10% N bases, low-quality sequences (no more than 1%), and adapter-contaminated sequences were removed. After this stringent filtering process, over 87% of the original reads were retained as clean reads (Appendix A), indicating a high level of data integrity suitable for subsequent analyses. To identify miRNAs from these clean reads, small RNA (sRNA) reads within the range of 18 to 35 nucleotides were extracted from the clean reads of each sample. The unique sRNA reads were denoted as “uniq”, while the total count was represented by “total”. The number and base composition of the selected sRNA reads were statistically analyzed, as detailed in Appendix A.

#### 2.5.2. Read Mapping to the Reference Genome

The reference genome AaegL5 was obtained from https://vectorbase.org/common/downloads/Current_Release/AaegyptiLVP_AGWG/fasta/data/ (VectorBase 68 accessed on 22 April 2024). Small RNAs with lengths ranging from 18 to 40 nucleotides were extracted from the raw data and aligned to the reference genome using Bowtie 1.0 [35], a software tool designed for aligning sequencing reads to long reference sequences. The expression levels and genomic distribution of these sRNAs were subsequently analyzed (Appendix A). The classification and further analysis of the sRNAs were based on the reads that successfully mapped to the reference genome.

#### 2.5.3. Differential Expression of miRNA

The input data for miRNA differential expression analysis consisted of readcount data obtained during miRNA expression level assessment. Given that both the ds*SHMT* and ds*EGFP* samples included three biological replicates, we employed DESeq2, which is based on the negative binomial distribution [36], to identify differentially expressed miRNAs. The screening criteria for differentially expressed miRNAs were set as follows: DESeq2 *p*-value ≤ 0.05 and |log_2_(fold change)| ≥ 0.2.

#### 2.5.4. GO and KEGG Enrichment Analysis

Gene Ontology (GO) enrichment analysis [37] was conducted on the target gene candidates of DEMs. Specifically, GOseq, which employs the Wallenius non-central hypergeometric distribution [38], was utilized to adjust for gene length bias in the GO enrichment analysis. The Kyoto Encyclopedia of Genes and Genomes (KEGG) [39] serves as a database resource for comprehending high-level functions and utilities of biological systems. To assess the statistical enrichment of the target gene candidates in KEGG pathways, we employed the KOBAS software 1.0 [40].

### 2.6. miRNA Antagomir and Mimic Treatment

Antagomirs and mimics of miRNAs were purchased from Dharmacon (Lafayette, CO, USA). The mimic of miRNA is a chemically synthesized mimic of miRNA, presented as a double-stranded molecule. The antagomirs were designed using the RNA module for custom single-stranded RNA synthesis as previously described [25]. When dissolved in nuclease-free water, 50 pmol of antagomir or mimic (100 μM, 0.5 μL per insect) was microinjected into the thorax of each cold-anesthetized female mosquito at 20–24 h post-eclosion (PE). The mosquitoes were then allowed a 3-day recovery period prior to blood feeding.

### 2.7. Statistical Analysis

All statistical analyses were conducted using GraphPad Prism 8 (GraphPad Software, San Diego, CA, USA), with data presented as mean ± standard error of the mean (SEM). For the RT-qPCR assay, each experiment was independently replicated a minimum of three times, with each replication comprising at least three technical replicates. Error bars represent variability among experimental replicates. The mean cycle threshold (Ct) values of RT-qPCR assay were transformed into relative expression levels through application of the 2^−ΔΔCt^ method [41,42,43]. Comparative analysis of expression levels was performed using a two-tailed unpaired Student’s t-test. Statistical significance thresholds were established as follows: * *p* < 0.05, ** *p* < 0.01, *** *p* < 0.001, **** *p* < 0.0001, and ns indicates “not significant” (*p* > 0.05).

## 3. Results

### 3.1. SHMT-Mediated miRNAs Revealed via Small RNA Sequencing Analysis

To investigate the impact of SHMT downregulation on miRNA expression, we compared the ds*SHMT* group and the ds*EGFP* control group at two time points: before and post-blood feeding. Prior to collecting samples for small RNA sequencing, it is essential to confirm *SHMT* downregulation and mosquito phenotypes. At 72 h post-injection of ds*SHMT* (72 h PIJ, before blood feeding), no significant differences were observed compared to the ds*EGFP* control group (Figure 1A). At 15 h post-blood meal (15 h PBM), mosquitoes in the ds*EGFP* group exhibited white abdominal patches and initiated ovarian development, whereas those with *SHMT* downregulation did not display these phenotypic changes (Figure 1B). Ovarian morphology showed no significant difference between groups at 72 h PIJ (Figure 1C), but ovarian development was markedly inhibited at 15 h PBM in the ds*SHMT* group (Figure 1D). RT-qPCR analysis confirmed that *SHMT* expression was significantly reduced in mosquitoes exhibiting abnormal phenotypes at 15 h PBM (Figure 1E). These findings align with our previous studies [34]. Consequently, we selected two time points—72 h PIJ (pre-blood feeding) and 15 h PBM—for small RNA sequencing.

Analysis indicated a predominance of sRNA sequences with lengths of 21–23 nucleotides, peaking at 22 nucleotides both before and after blood feeding, which is consistent with the characteristic length distribution of miRNAs (Figure 2A–D). A comprehensive statistical analysis of the diversity and abundance of the selected sRNA sequences was performed (Appendix A), revealing that approximately 90% of the sequence lengths in each sample were successfully mapped to the *Ae. aegypti* genome (Appendix A). After stringent quality control of the raw sequencing data and subsequent selection of sRNA sequences, a total of 220 miRNAs were annotated or identified, including 151 known miRNAs from *Ae. aegypti* and 69 novel miRNAs (Appendix A).

Prior to blood feeding, a comparative analysis between the dsSHMT group (B_dsS) and the control dsEGFP group identified differential expression of 20 miRNAs (*p*-value ≤ 0.05 and |log_2_(foldchange)| ≥ 0.2). Among these, 12 miRNAs (aae-miR-10365, miR-11894a, miR-11898, miR-13-3p, miR-13-5p, miR-14, miR-2940-3p, miR-2940-5p, miR-305-5p, miR-315-5p, miR-957, and novel_139) were upregulated, while 8 miRNAs (aae-miR-1, miR-100, miR-278-3p, miR-279, miR-281-5p, miR-2941, miR-2946, and novel_22) were downregulated in the ds*SHMT* group (Figure 3A; Table 1). Following blood meal ingestion, six miRNAs (aae-miR-1174, miR-2940-3p, miR-2940-5p, miR-2b, miR-14, and novel_55) were upregulated, and seven miRNAs (aae-miR-11-5p, miR-252-5p, miR-278-5p, miR-305-5p, miR-8-5p, novel_22, and novel_47) were downregulated in the ds*SHMT* group (P_dsS) (Figure 3B; Table 2). Notably, miR-2940-3p, miR-2940-5p, miR-305-5p, and novel_22 exhibited the most pronounced responses to *SHMT* depletion in mosquitoes both before and after blood feeding (*p*-adj ≤ 0.01 and |log_2_(foldchange)| ≥ 0.2). Intriguingly, miR-305-5p demonstrated an inverse response pattern, being upregulated prior to blood feeding and downregulated afterward.

### 3.2. Validation of DEMs at 72 H PIJ Through Real-Time Quantitative PCR

The DEMs identified through high-throughput sequencing were subsequently validated using real-time quantitative PCR (RT-qPCR). The results demonstrated a significant upregulation of ten miRNAs (aae-miR-1174, miR-1175, miR-2940-3p, miR-2940-5p, miR-305-5p, let-7, miR-34-5p, miR-8-3p, miR-306-5p, and miR-184) in the ds*SHMT* group compared to both the wild-type (WT) and ds*EGFP* groups (Figure 4). Conversely, three miRNAs (aae-miR-263a-5p, miR-286b, and miR-2941) showed marked downregulation (Figure 4). Additionally, seven miRNAs (aae-miR-9a, miR-279, miR-309a, miR-375, miR-989, miR-2944b-5p, and miR-2946) exhibited no significant changes in expression levels (Figure 4). Collectively, these qPCR validation results corroborate the sequencing data, thereby confirming the differential expression patterns of these miRNAs.

### 3.3. Validation of DEMs at 15 H PBM Through Real-Time Quantitative PCR

RT-qPCR was also employed to validate the DEMs identified through sequencing at 15 h PBM (Figure 5). The results indicated that seven miRNAs (aae-miR-1174, miR-1175, miR-2940-3p, miR-2940-5p, miR-306-5p, let-7, and miR-34-5p) were significantly upregulated. Notably, miR-305-5p exhibited an upregulation at 72 h PIJ (before blood meal) but was downregulated at 15 h PBM. Additionally, twelve miRNAs (aae-miR-8-3p, miR-184, miR-263a-5p, miR-286b, miR-2941, miR-9a, miR-279, miR-309a, miR-375, miR-989, miR-2944b-5p, and miR-2946) showed no significant changes in expression in response to *SHMT* RNAi. In summary, the qPCR validation results were consistent with the sequencing data obtained at 15 h PBM.

### 3.4. Biological Roles of Differentially Expressed microRNAs

Subsequently, we administered miRNA mimics and antagomirs to investigate the functional roles of four miRNAs (miR-2940-5p, miR-2940-3p, miR-2941, and miR-306-5p) that exhibited differential expression in response to *SHMT* deletion, both prior to and following blood feeding. Downregulation of miR-2940-3p did not impact blood digestion in adult mosquitoes compared to the wild-type (WT) and control group Ant-67 (Figure 6A); however, it did result in less developed ovaries at 48 h PBM (Figure 6B). Conversely, the overexpression of miR-2940-3p did not result in any discernible alterations in the phenotypic appearance of adult mosquitoes (Figure 6C). Upon dissection, no significant differences were observed in midgut blood digestion or crop sugar water storage (Figure 6C). However, compared to the control groups, the injection of miR-2940-3p mimic resulted in slightly larger size observed at the same time point (Figure 6D). The downregulation of miR-2940-3p was confirmed by RT-qPCR analysis (Figure 6E). Notably, the reduction in miR-2940-3p expression did not influence mosquito blood feeding rate (Figure 6F), but it significantly impacted oviposition (Figure 6G). In contrast, the downregulation of miR-2940-5p, miR-2941, and miR-306-5p had no significant effect on oviposition (Figure 6G). Overexpression of miR-2940-3p was validated through RT-qPCR analysis (Figure 6H). Statistical analysis revealed that the upregulation of these four miRNAs did not result in a significant alteration in the blood feeding rate of mosquitoes (Figure 6I) or in their egg-laying rate (Figure 6J).

### 3.5. SHMT Knockdown Influences the miRNA Biosynthesis Pathway

The production and functional mechanisms of miRNAs necessitate the involvement of a series of nucleases, including Drosha, Dicer, and AGO [44]. We hypothesize that the differential expression of miRNAs resulting from the downregulation of *SHMT* may be associated with alterations in the expression levels of these nucleases. To test this hypothesis, we utilized RT-qPCR to evaluate the expression levels of these enzymes in *SHMT*-depleted mosquitoes at 72 h PIJ (before blood meal) and 15 h PBM. In mosquitoes at both time points, the transcriptional expression of *Drosha*, a key enzyme responsible for converting pri-miRNA to pre-miRNA, was significantly reduced compared to both the wild-type (WT) and ds*EGFP* control groups (Figure 7A,B). Similarly, the expression of *Dicer1*, which is essential for the maturation of double-stranded miRNAs, was also significantly decreased (Figure 7A,B). Conversely, the transcriptional expression of *Dicer2*, involved in siRNA biogenesis, was significantly elevated (Figure 7A,B). Additionally, the expression levels of *AGO1* and *AGO2*, both integral components in the assembly of the RNA-induced silencing complex (RISC), were significantly downregulated (Figure 7A,B). Notably, no significant changes were observed in the expression levels of AGO3 (Figure 7A,B). The downregulation of *SHMT* appears to modulate the expression of specific miRNAs, potentially through its interaction with key enzymes within the miRNA processing pathway. Notably, the selective impact of *SHMT* downregulation on a limited subset of miRNAs suggests a targeted regulatory mechanism. Furthermore, while transcriptional alterations in genes encoding these miRNA-processing enzymes were observed following *SHMT* downregulation, the corresponding changes at the protein level remain to be elucidated. These findings underscore the need for further investigation into the precise molecular mechanisms underlying the selective influence of *SHMT* on miRNA expression, as well as the nature of the regulatory interplay between *SHMT* and these miRNAs.

## 4. Discussion

Emerging evidence has demonstrated that the biological role of SHMT in mammalian and insect systems is closely associated with distinct classes of non-coding RNAs, with a particular emphasis on miRNAs. Notably, SHMT has been identified as a target gene for several miRNAs, highlighting its regulatory complexity. Studies on lung adenocarcinoma have shown that miR-218-5p reduces the cytotoxic activity of natural killer cells against lung adenocarcinoma cells by targeting *SHMT1* [45]. *SHMT2* has been found to harbor binding sites for miR-370, with an inverse relationship observed in the context of cartilage degradation [46]. miR-615-5p inhibits cell proliferation and migration by exerting a negative regulatory effect on *SHMT2* in hepatocellular carcinoma [47,48]. The development of gastric cancer is associated with the upregulation of *SHMT2* mediated by miR-449a, while circ_0063526 enhances drug resistance in gastric cancer through modulation of the miR-449a/*SHMT2* axis [49]. We have demonstrated that mosquito- and gut-specific miRNA, miR-1174, modulates the biological functions of the mosquito midgut by targeting *SHMT* [25]. Further studies showed that depletion of *SHMT* results in a spectrum of abnormal phenotypes, such as reduced flight capacity, impeded blood digestion, suboptimal ovarian development, and a reduction in egg-laying [34]. It is important to highlight that our previous research has demonstrated that *SHMT* exhibits high expression levels in mosquito tissues outside the midgut, whereas its expression within the midgut is markedly diminished. This contrasts sharply with the abundance of digestive enzymes present in this region [34]. We propose that the biological dysfunction of the mosquito midgut resulting from *SHMT* downregulation may be indirectly mediated through its influence on downstream gene expression regulatory networks or key molecular pathways. Specifically, alterations in *SHMT* expression could impact critical metabolic pathways such as carbon and amino acid metabolism, thereby modifying intracellular methylation status and epigenetic regulation, ultimately leading to midgut functional abnormalities. While prior studies have indicated that *SHMT* expression is regulated post-transcriptionally by specific miRNAs, there is currently no direct evidence demonstrating a feedback regulatory effect of *SHMT* on miRNA expression. Clarifying this scientific question has significant implications for advancing our understanding of the molecular mechanisms underlying SHMT’s role in regulating mosquito midgut function.

In total, 20 DEMs were identified in response to *SHMT* downregulation before or post-blood meal (Figure 3). Following this, functional studies were conducted on four *SHMT*-responsive miRNAs by injecting miRNA mimics and antagomirs. The results indicated that miR-2941, miR-2940-5p, and miR-306-5p did not affect blood digestion and absorption or ovarian development; nor did they cause significant changes in abdominal distension rate, blood feeding rate, blood meal allocation, or egg-laying rate (Figure 6). However, the downregulation of miR-2940-3p alone resulted in reduced ovarian development (Figure 6), consistent with the previous results [50]. We acknowledge that the functional analysis of these DEMs in this study is merely a preliminary exploration. The absence of observable phenotypic abnormalities in mosquitoes following upregulation or downregulation does not imply their insignificance. This phenotypic resilience could potentially be attributed to multiple underlying factors: (1) the magnitude of miRNA expression alteration might have been insufficient to elicit detectable phenotypic changes; (2) these miRNAs may exert subtle regulatory functions that are not readily apparent at the organismal level; and (3) potential activation of compensatory molecular pathways within the organism may have mitigated the effects of miRNA modulation. A comparable scenario was observed in our previous studies of silkworms. Neither antagomirs nor mimics of miRNAs exhibit significant effects in the silkworm silk glands [51]. Only the knockout and transgenic overexpression of let-7 miRNAs resulted in observable phenotypic abnormalities [52]. Considering the highly specific and complex nature of miRNA functions, it is crucial to employ advanced investigative approaches, particularly gene editing technologies, to further elucidate their biological roles.

The biogenesis and expression of miRNAs are tightly regulated by diverse factors, with specific nucleases in nuclear and cytoplasmic compartments driving critical steps of the maturation cascade. The pri-miRNA and pre-miRNA are transcribed in the nucleus [44], transported from the nucleus to the cytoplasm via the Exportin-5 protein [53], and ultimately processed into mature miRNA in the cytoplasm [54]. Within the context of miRNA biogenesis, Drosha facilitates the cleavage of pri-miRNAs into precursor miRNAs (pre-miRNAs) that possess a stem-loop structure of approximately 70 nucleotides in length. These pre-miRNAs are subsequently exported from the nucleus with the aid of the transport protein Exportin 5. In the cytoplasm, they undergo further processing by the *Dicer* enzyme, resulting in the formation of mature miRNAs, which are double-stranded structures of approximately 22 nucleotides in length. In miRNA maturation, the *AGO* and *Dicer* enzymes coalesce to form the RNA-induced silencing complex (RISC), facilitating the selective degradation of one strand within the miRNA duplex to yield a mature single-stranded miRNA [55]. Disruption of *SHMT* expression leads to the downregulation of *Drosha*, *Dicer1*, and *AGO1*, while upregulating the expression of *Dicer2* and *AGO2* (Figure 7). These findings suggest that *SHMT* potentially modulates miRNA expression by influencing key enzymes involved in the miRNA biogenesis pathway. However, among the miRNAs identified in this study, only a subset exhibited significant expression changes, including both upregulated and downregulated species. This selective pattern of miRNA regulation indicates that *SHMT* may specifically target certain miRNAs rather than exerting a global effect on miRNA expression. The production and expression of these *SHMT*-responsive miRNAs are likely influenced by additional regulatory factors, suggesting a complex regulatory network. Apart from influencing the enzymes involved in the miRNA biogenesis process, alterations in *SHMT* expression may activate or inhibit certain signaling pathways, such as the PI3K/AKT intracellular signaling pathway [56]. The activation or inhibition of the PI3K/AKT signaling pathway may, in turn, affect the expression of miRNAs [57]. Furthermore, downregulation of *SHMT* can alter the levels of intracellular methyl donors [58] and nucleotide synthesis precursors [59], and the reduction in methyl donors may impact the modification of miRNA precursors, thereby affecting their stability or maturation efficiency [60,61].

The precise molecular mechanisms underlying *SHMT*-mediated regulation of specific miRNAs warrant further comprehensive investigation. Future research could focus on the following directions: first, further employing RNA interference technology combined with high-throughput sequencing to systematically analyze changes in miRNA expression profiles upon specific downregulation of SHMT across various mosquito species; second, utilizing chromatin immunoprecipitation (ChIP) to investigate whether SHMT plays a role in regulating epigenetic modifications of miRNA promoter regions; finally, verifying the regulatory relationship between SHMT and miRNA in vivo through the construction of transgenic mosquito models with SHMT overexpression and knockout. These studies will contribute to unraveling the molecular mechanisms of SHMT in regulating mosquito midgut function and provide a theoretical foundation for developing novel strategies for the prevention and control of mosquito-borne infectious diseases.

## 5. Conclusions

The midgut physiology of *SHMT*-deficient mosquitoes exhibits dysfunction, characterized by aberrant expression of digestive enzymes and impaired blood digestion. The underlying mechanism by which *SHMT* regulates this abnormal phenotype in mosquitoes remains elusive. In this study, we identified DEMs in response to *SHMT* downregulation by small RNA sequencing and RT-qPCR. We further investigated selected DEMs, including miR-2940-5p, miR-2940-3p, miR-2941, and miR-306-5p, conducting preliminary studies on their biological functions. To elucidate the molecular mechanisms of miRNA responses to *SHMT* downregulation, we examined the expression levels of genes involved in miRNA biogenesis pathways. Our findings indicate that several key enzymes, such as Drosha, Dicer1, and AGO1, are downregulated in response to *SHMT* downregulation, while Dicer2 and AGO2 exhibit upregulation. These results provide novel insights into the molecular mechanisms by which *SHMT* modulates midgut biological function and nutrient metabolism in mosquitoes.

## Figures and Tables

**Figure 1 biomolecules-15-00644-f001:**
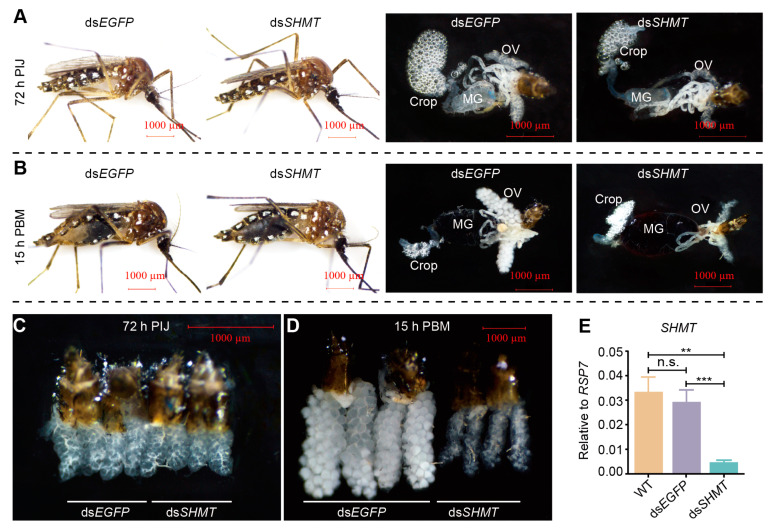
Mosquitoes injected with ds*SHMT* and ds*EGFP* were evaluated both prior to blood feeding (72 h post-injection, 72 h PIJ) and 15 h post-blood meal (15 h PBM). (**A**) Individual and anatomical images of mosquitoes at 72 h PIJ. (**B**) Individual and anatomical images of mosquitoes at 15 h PBM. (**C**) Ovaries of mosquitoes at 72 h PIJ. (**D**) Ovaries of mosquitoes at 15 h PBM. (**E**) The downregulation of *SHMT* was confirmed through RT-qPCR at 15 h PBM. *RPS7* was used as the control for RT-qPCR. Each experimental group included a minimum of three biological replicates, with each biological replicate comprising three technical replicates. Statistical analyses were conducted using one-way ANOVA followed by unpaired two-tailed Student’s *t*-test. Data are presented as mean ± SEM from at least three independent experiments (n ≥ 3 per group). MG, midgut; OV, ovary; WT, wild *Ae. aegypti* without any treatment; ds*EGFP*, double-strand RNA of EGFP; ds*SHMT*, double-strand RNA of *SHMT*. ** *p* < 0.01, *** *p* < 0.001, n.s., not significant. The scale bar value in all images of (**A**–**D**) is 1000 μm.

**Figure 2 biomolecules-15-00644-f002:**
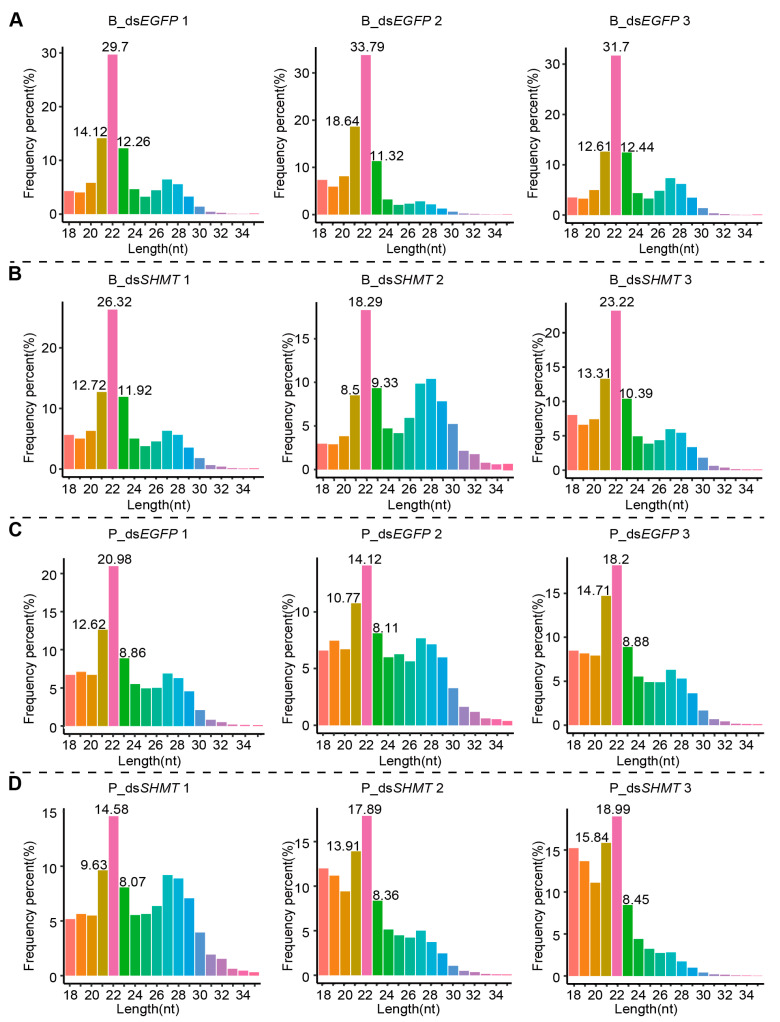
Raw data quality control and sRNA screening. (**A**) Length distribution of total clean reads of B_ds*EGFP*. (**B**) Length distribution of total clean reads of B_ds*SHMT*. (**C**) Length distribution of total clean reads of P_ds*EGFP*. (**D**) Length distribution of total clean reads of P_ds*SHMT*. There were three biological replicates in both the experimental group (ds*SHMT*) and the control group (dsEGFP), with each replicate consisting of total RNA or small RNA extracted from six mosquitoes. Each column represents an individual biological replicate. In the sample labels, “B” indicates samples collected before blood meal at 72 h post-injection (72 h PIJ), while “P” indicates samples collected at 15 h after blood feeding (15 h PBM). B_ds*EGFP* and B_ds*SHMT* represent the samples collected before blood feeding, with B_ds*SHMT* being the treated group and B_ds*EGFP* being the control group. P_ds*EGFP* and P_ds*SHMT* represent the samples collected after blood feeding, with P_ds*SHMT* being the treated group and P_ds*EGFP* being the control group.

**Figure 3 biomolecules-15-00644-f003:**
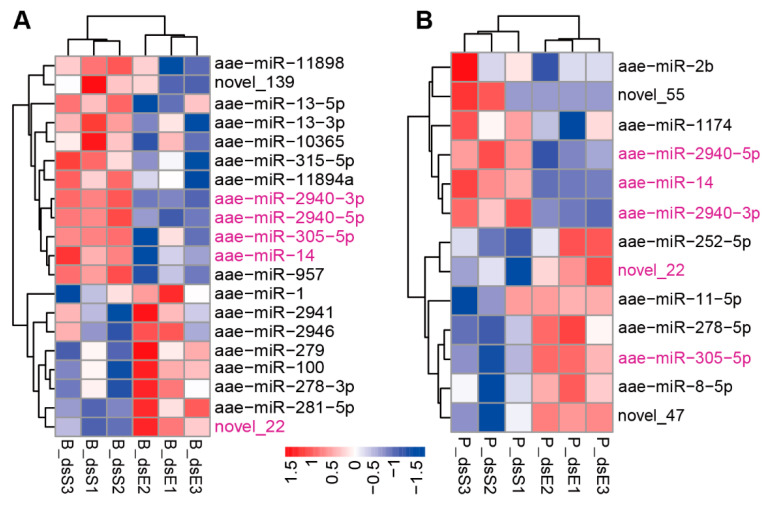
Differentially expressed miRNAs (DEMs). (**A**) Clustering analysis of DEMs at 72 h PIJ (72 h post-injection). (**B**) Clustering analysis of DEMs at 15 h PBM (15 h post-blood meal). There were three biological replicates in both the experimental group (ds*SHMT*, ds*S*) and the control group (ds*EGFP*, ds*E*), with each replicate consisting of total RNA or small RNA extracted from eight mosquitoes. The numbers 1, 2, and 3 in the heatmap represent three biological replicates. In the sample labels, “B” indicates samples collected before the blood meal at 72 h post-injection (72 h PIJ), while “P” indicates samples collected at 15 h after blood feeding (15 h PBM). B_ds*E* and B_ds*S* represent the samples collected before blood feeding, with B_ds*S* being the treated group and B_ds*E* being the control group. P_ds*E* and P_ds*S* represent the samples collected after blood feeding, with P_ds*S* being the treated group and P_ds*E* being the control group. The screening criteria for differentially expressed miRNAs were set as follows: *p*-value ≤ 0.05 and |log_2_(fold change)| ≥ 0.2. Red indicates high expression, while blue indicates low expression. The bar gradient from red to blue represents the range of log_10_(TPM + 1) values from high to low.

**Figure 4 biomolecules-15-00644-f004:**
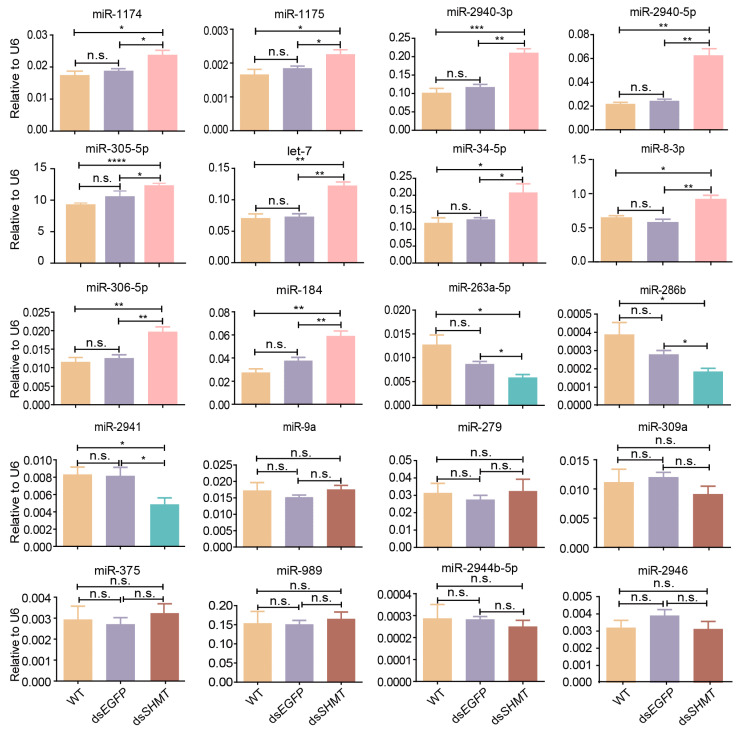
Quantitative PCR validation of miRNAs in mosquitoes at 72 h post-injection (prior to blood feeding). Each experimental group included a minimum of three biological replicates, with each biological replicate comprising three technical replicates. The small nuclear RNA (snRNA) *U6* was used as the control for RT-qPCR. Statistical analyses were conducted using one-way ANOVA followed by unpaired two-tailed Student’s *t*-test. Data are presented as mean ± SEM from at least three independent experiments (n ≥ 3 per group). WT, wild *Ae. aegypti* without any treatment; ds*EGFP*, double-strand RNA of *EGFP*; ds*SHMT*, double-strand RNA of *SHMT*. * *p* < 0.05, ** *p* < 0.01, *** *p* < 0.001, **** *p* < 0.0001, n.s., not significant.

**Figure 5 biomolecules-15-00644-f005:**
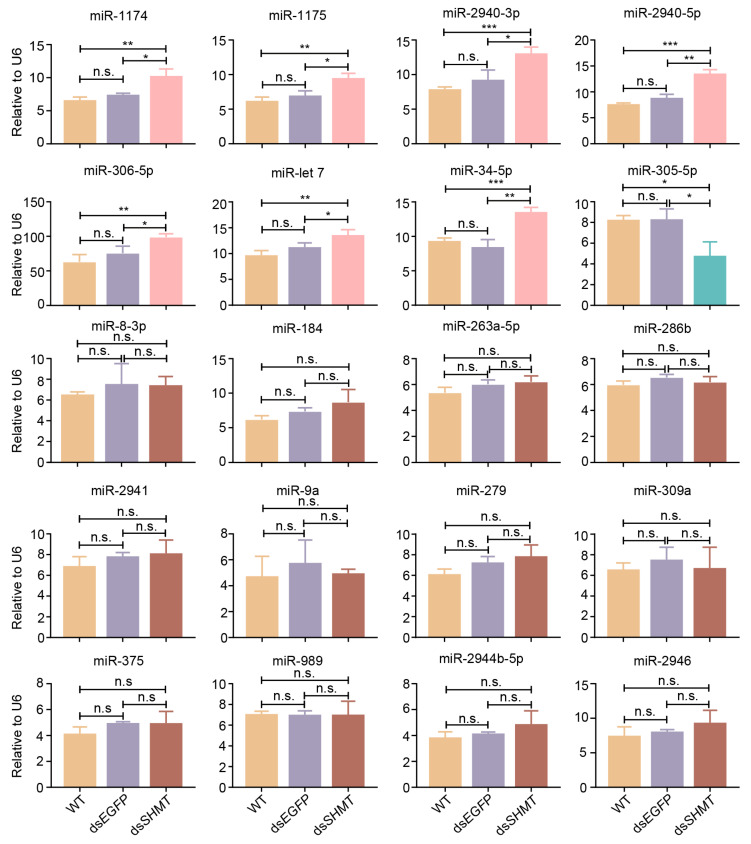
RT-qPCR verification of differentially expressed miRNAs and non-responsive miRNAs after downregulation with *SHMT* at 15 h post-blood meal (15 h PBM). The small nuclear RNA (snRNA) *U6* was used as the control for RT-qPCR. Statistical analyses were conducted using one-way ANOVA followed by unpaired two-tailed Student’s *t*-test. Data are presented as mean ± SEM from at least three independent experiments (n ≥ 3 per group). WT, wild *Ae. aegypti* without any treatment; ds*EGFP*, double-strand RNA of *EGFP*; ds*SHMT*, double-strand RNA of *SHMT*. * *p* < 0.05, ** *p* < 0.01, *** *p* < 0.001, n.s., not significant.

**Figure 6 biomolecules-15-00644-f006:**
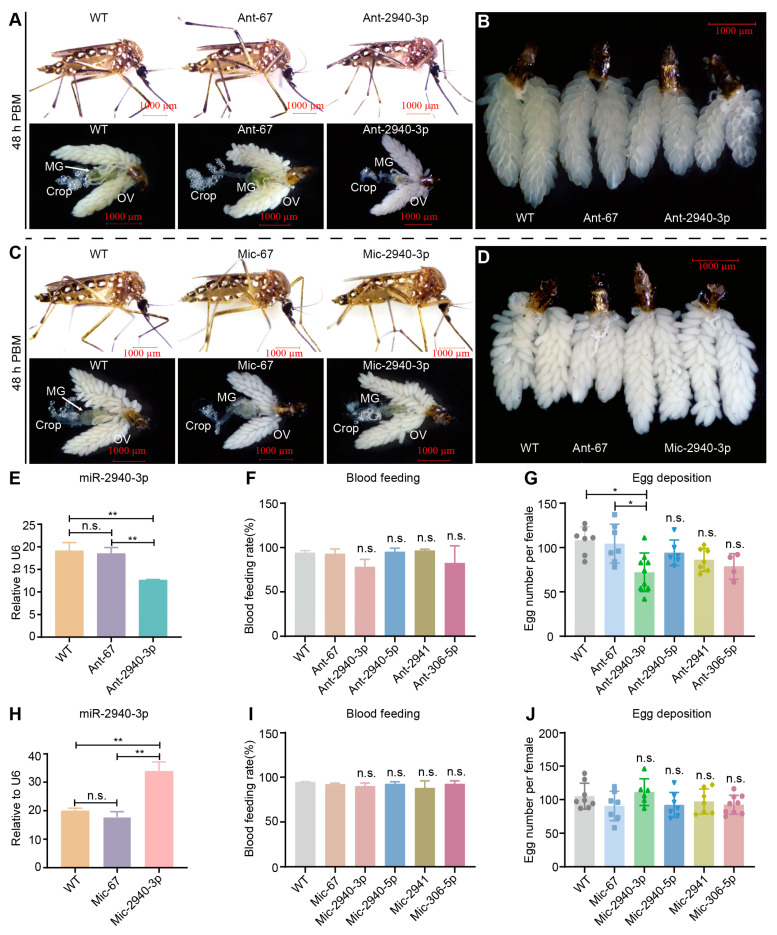
Statistics of abnormal phenotypes for Ant-2940-3p and Mic-2940-3p. (**A**) Abnormal phenotypes resulted from Ant-2940-3p at 48 h PBM. (**B**) Ovary phenotypes by Ant-2940-3p at 48 h PBM. (**C**) Abnormal phenotypes resulted from Mic-2940-3p at 48 h PBM. (**D**) Ovary phenotypes caused by Mic-2940-3p at 48 h PBM. (**E**) Knockdown miR-2940-3p by Ant-2940-3p. (**F**) Statistics of blood feeding rate after injection of Ant-2940-3p. (**G**) Laying rate for Ant-2940-3p. (**H**) Overexpression of miR-2940-3p by Mic-2940-3p. (**I**) Statistics of blood feeding rate after injection of Mic-2940-3p. (**J**) Laying rate for Mic-2940-3p. The small nuclear RNA (snRNA) *U6* was used as the control for RT-qPCR. Each group included at least three biological replicates, with each biological replicate comprising three technical replicates. Statistical analyses were conducted using one-way ANOVA followed by unpaired two-tailed Student’s *t*-test. Data are presented as mean ± SEM from at least three independent experiments (n ≥ 3 per group). PBM, post-blood meal; WT, wild *Ae. aegypti* without any treatment; Mic, the mimic of miRNA; Ant, the antagomir of miRNA. * *p* < 0.05, ** *p* < 0.01, n.s., not significant. The scale bar value in all images of (**A**–**D**) is 1000 μm.

**Figure 7 biomolecules-15-00644-f007:**
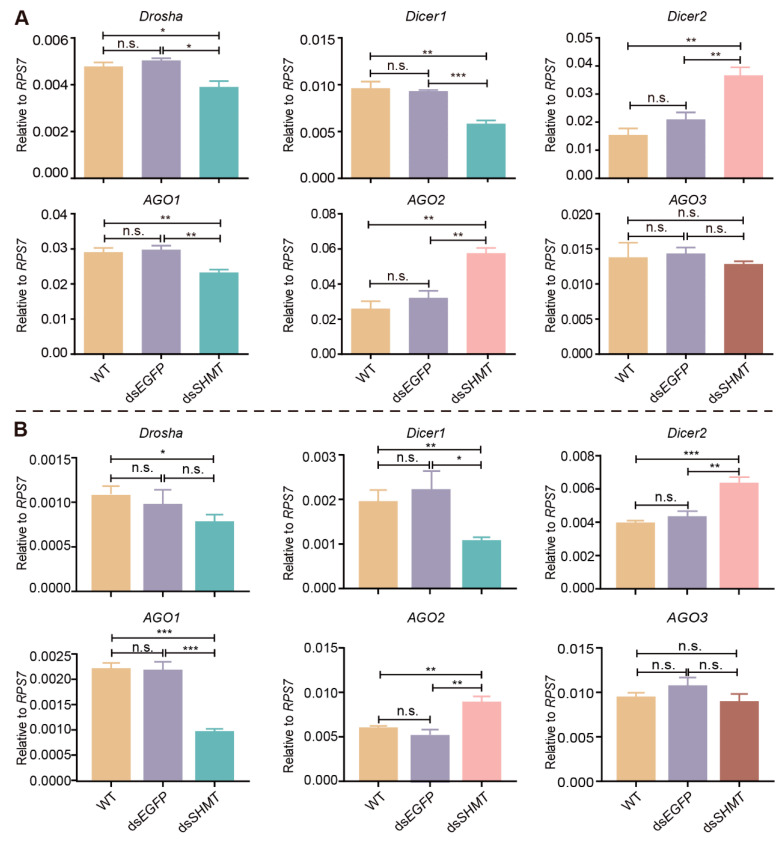
Quantitative PCR analysis of key enzymes involved in miRNA biogenesis pathways. (**A**) RT-qPCR assay at 72 h post-injection (72 h PIJ, before blood meal); (**B**) RT-qPCR assay at 15 h post-blood meal (15 h PBM). Each experimental group included a minimum of three biological replicates, with each biological replicate comprising three technical replicates. *RPS7* was used as the control for RT-qPCR. Statistical analyses were conducted using one-way ANOVA followed by unpaired two-tailed Student’s *t*-test. Data are presented as mean ± SEM from at least three independent experiments (n ≥ 3 per group). WT, wild *Ae. aegypti* without any treatment; ds*EGFP*, double-strand RNA of *EGFP*; ds*SHMT*, double-strand RNA of *SHMT*. * *p* < 0.05, ** *p* < 0.01, *** *p* < 0.001, n.s., not significant.

**Table 1 biomolecules-15-00644-t001:** SHMT-responsive miRNAs in pre-bloodsucking mosquitoes at 72 h post-injection.

miRNA	B_dsS_Readcount	B_dsE_Readcount	Log_2_foldchange	*p*-Val	*p*-Adj	Response
aae-miR-2940-3p	38,503.96064	17,963.88509	1.099846276	1.27 × 10^−22^	2.75 × 10^−20^	up
aae-miR-2940-5p	28,776.17523	14,239.18183	1.015030016	1.31 × 10^−15^	1.42 × 10^−13^	up
aae-miR-10365	1672.989849	1303.784612	0.358497771	0.038818301	0.479559669	up
aae-miR-11894a	1222.489078	887.4538831	0.46232103	0.003855015	0.118954763	up
aae-miR-11898	41.06098255	21.5461849	0.95894345	0.021603411	0.40864567	up
aae-miR-13-3p	22,546.40415	18,133.37452	0.314187719	0.011490061	0.248185318	up
aae-miR-13-5p	349.6802822	246.7725908	0.496899458	0.024177837	0.40864567	up
aae-miR-14	74,719.97782	58,674.92568	0.348711384	0.008823778	0.211770679	up
aae-miR-305-5p	11,070.20678	6645.786433	0.735803189	5.85 × 10^−5^	0.003160797	up
aae-miR-315-5p	15,999.77039	13,614.42772	0.232868934	0.040695943	0.479559669	up
aae-miR-957	13,867.09344	11,558.02215	0.262582036	0.024594415	0.40864567	up
aae-miR-1	372,756.3313	436,433.1172	−0.227531884	0.046939287	0.479559669	down
aae-miR-100	226,244.2959	263,694.4811	−0.22098482	0.033555478	0.472595793	down
aae-miR-278-3p	5759.313951	7506.544882	−0.382049339	0.001315148	0.056814412	down
aae-miR-279	15,401.80793	19,926.74112	−0.371521561	0.002496829	0.089885837	down
aae-miR-281-5p	80,276.02829	100,106.3372	−0.318492468	0.004670184	0.126094959	down
aae-miR-2941	6771.211175	8089.190071	−0.256322377	0.048873163	0.479559669	down
aae-miR-2946	2113.090291	2587.873342	−0.292018987	0.035007096	0.472595793	down
novel_22	73.83129566	166.6737009	−1.167479519	1.76 × 10^−6^	0.000126948	down

Note: B_dsS_readcount and B_dsE_readcount are the normalized readcounts for B_dsS and B_dsE, respectively.

**Table 2 biomolecules-15-00644-t002:** SHMT-responsive miRNAs in mosquitoes at 15 h post-blood meal.

miRNA	P_dsS_Readcount	P_dse_Readcount	Log_2_foldchange	*p*-Val	*p*-Adj	Response
aae-miR-11-5p	1664.66266	2356.000046	−0.498733975	0.041077109	0.680571033	up
aae-miR-1174	6353.65955	4130.419729	0.621354561	0.013352252	0.299683875	up
aae-miR-14	1,9701.04808	15,953.54525	0.30438114	0.003283273	0.132644239	up
aae-miR-2940-3p	10,621.09158	7665.246502	0.470924008	9.40 × 10^−6^	0.000949497	up
aae-miR-2940-5p	9800.348606	7564.805556	0.373575047	0.005986621	0.201549588	up
aae-miR-2b	2862.942607	2230.901831	0.35811565	0.043799126	0.680571033	up
novel_55	4.104653461	0	4.622962489	0.031714775	0.64063846	up
aae-miR-252-5p	3330.881667	3923.49034	−0.236519613	0.036052746	0.662059515	down
aae-miR-278-5p	125.7710599	185.1747529	−0.547435742	0.009131322	0.239686085	down
aae-miR-305-5p	15,072.75569	26176.11006	−0.796115398	1.59 × 10^−9^	3.20 × 10^−7^	down
aae-miR-8-5p	4121.121114	4987.051375	−0.27482532	0.009492518	0.239686085	down
novel_22	363.33811	595.7105685	−0.717321745	0.00058874	0.0396418	down
novel_47	36.58597513	72.55578887	−0.957760648	0.002476551	0.125065816	down

Note: B_dsS_readcount and B_dsE_readcount are the normalized readcount for B_dsS and B_dsE, respectively.

## Data Availability

The data supporting this article have been included as part of the main figures or as Appendix A. Further inquiries can be directed to the corresponding author.

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
