# Peer review of "Serine Hydroxymethyltransferase Modulates Midgut Physiology in Aedes aegypti Through miRNA Regulation: Insights from Small RNA Sequencing and Gene Expression Analysis"

_biomolecules, 2025, doi:10.3390/biom15050644_

Round 1
Reviewer 1 Report
Comments and Suggestions for Authors
The article - Serine hydroxymethyltransferase-mediated microRNAs in Aedes aegypti mosquitoes by Pu et al., suggests that several key enzymes, including Drosha, Dicer1, and AGO1, involved in the miRNA biogenesis pathway, exhibited altered expression in response to SHMT downregulation. Overall, I enjoyed reading the manuscript and think that it has useful information related to SHMT influenced differential expression of key enzymes involved in the miRNA biogenesis pathway. However, there are certain critical issues that need to be addressed, which are as follows:
Title: The title “Serine hydroxymethyltransferase-mediated microRNAs in Ae- 2
des aegypti mosquitoes” seems to be incomplete and should be changed.
Line 30: Restructure the introduction with an overview of Aedes aegypti biology and miRNA function before focusing on SHMT to highlight the connections between SHMT, blood digestion, and miRNA regulation.
Line 61: Expand this section to consider whether metabolic stress or other indirect physiological factors might be contributing to the observed effects.
Line 149: Update the reference genome information (AaegL5) to https://vectorbase.org/common/downloads/Current_Release/AaegyptiLVP_AGWG/fasta/data/.
Line 158: I am not sure if the authors meant p-adj by “corrected p-values.” The data (DESeq2 results) should be analyzed using adjusted p-values (p-adj) rather than raw p-values. Also, the authors should clarify whether they have applied a fold-change cutoff to define significant changes; if so, specify that cutoff value along with the FDR threshold. The authors should also mention the RINs DESeq2 results for the samples used for sequencing and the cut-off used for count data. Explicitly stating these criteria will strengthen the analysis’s statistical rigor. Also, the data should be made available as supplemental data.
Line 195: Provide a rationale for selecting 72 h PIJ and 15 h PBM as experimental time points. Explain why these intervals are biologically relevant.
Line 240: Clearly define the criteria for identifying differentially expressed miRNAs (DEMs), including the fold-change cutoff and FDR threshold applied.
Line 327: Discuss whether the observed changes might suggest for a compensatory mechanism.
Line 387: It is highly surprising that, miR-1174 and miR-1175 are upregulated by the depletion of SHMT both at 72h PIJ and 15h PBM. One of the corresponding author’s previous articles published in PNAS, 2014, showed that miR-1174 targets SHMT to maintain a low SHMT level in the mosquito gut by direct action on the SHMT transcript. It is not clear how both can be true. The authors should explain this in detail.
Figure 1E: Replace RSP7 with RPS7.
Author Response
- The article - Serine hydroxymethyltransferase-mediated microRNAs in Aedes aegypti mosquitoes by Pu et al., suggests that several key enzymes, including Drosha, Dicer1, and AGO1, involved in the miRNA biogenesis pathway, exhibited altered expression in response to SHMT downregulation. Overall, I enjoyed reading the manuscript and think that it has useful information related to SHMT influenced differential expression of key enzymes involved in the miRNA biogenesis pathway. However, there are certain critical issues that need to be addressed, which are as follows:
Responding: We sincerely thank you for taking the time and effort to carefully review our manuscript amidst your busy schedule. Your opinions and suggestions are highly valuable and have significantly contributed to improving the quality of our manuscript. We have revised the manuscript according to your suggestions and provided a point-by-point response to your feedback.
- Title: The title “Serine hydroxymethyltransferase-mediated microRNAs in Aedes aegypti mosquitoes” seems to be incomplete and should be changed.
Responding: Thank you for this suggestion. The title has been changed to “Serine Hydroxymethyltransferase Modulates Midgut Physiol-ogy in Aedes aegypti through miRNA Regulation: Insights from Small RNA Sequencing and Gene Expression Analysis”.
- Line 30: Restructure the introduction with an overview of Aedes aegypti biology and miRNA function before focusing on SHMT to highlight the connections between SHMT, blood digestion, and miRNA regulation.
Responding: Thanks for your suggestion. We have made appropriate adjustments to the Introduction according to your suggestion. Please see lines 30-95.
- Line 61: Expand this section to consider whether metabolic stress or other indirect physiological factors might be contributing to the observed effects.
Responding: Thanks for this suggestion. We have rewritten this paragraph. Please see lines 51-70.
- Line 149: Update the reference genome information (AaegL5) to https://vectorbase.org/common/downloads/Current_Release/AaegyptiLVP_AGWG/fasta/data/.
Responding: Thanks for this suggestion. The reference genome information has been updated. Please see lines 177-178.
- Line 158: I am not sure if the authors meant p-adj by “corrected p-values.” The data (DESeq2 results) should be analyzed using adjusted p-values (p-adj) rather than raw p-values. Also, the authors should clarify whether they have applied a fold-change cutoff to define significant changes; if so, specify that cutoff value along with the FDR threshold. The authors should also mention the RINs DESeq2 results for the samples used for sequencing and the cut-off used for count data. Explicitly stating these criteria will strengthen the analysis’s statistical rigor. Also, the data should be made available as supplemental data.
Responding:Thank you very much for your comment and suggestion. We are aware that employing more stringent criteria, such as p-adj ≤ 0.05 and |log2(fold change)| ≥ 1, for screening differentially expressed miRNAs would likely yield more robust results. However, the number of differentially expressed miRNAs identified under these conditions was relatively limited. Consequently, we opted to relax the screening criteria to identify a broader range of potentially differentially expressed miRNAs. The screening parameters for differential expression were finally set as p-value ≤ 0.05 and |log2(fold change)| ≥ 0.2. Given the relatively lenient thresholds applied, we further validated the candidate miRNAs using quantitative PCR. Additionally, for ease of reference, two tables (Tables 1 and 2) were included to provide a clearer visualization of the screening results. These tables highlight the miRNAs exhibiting the most significant response to SHMT downregulation. Please see lines 185-190 and Tables 1&2 in the Result section.
- Line 195: Provide a rationale for selecting 72 h PIJ and 15 h PBM as experimental time points. Explain why these intervals are biologically relevant.
Responding:Thank you very much for your comment and suggestion. After the injection of double-stranded RNA, we planned to collect samples at two time points for sequencing and quantitative analysis: before and after blood feeding. Since the injection wounds were completely healed by 3 days post-injection (72h PI), and the mosquitoes resumed normal physiological activities, including the ability to feed on blood, we considered this time point suitable for sample collection to avoid interference from the injection trauma. Prior to blood feeding, mosquitoes with downregulated SHMT did not exhibit any visible abnormal phenotypes. However, at 15 hours post-blood meal (15h PBM), phenotypes such as delayed blood digestion and ovarian development began to emerge, suggesting that this time point is appropriate for detecting early molecular changes following blood feeding. Therefore, in this study, samples were collected at 72 hours post-injection (before blood feeding) and 15 hours post-blood meal for sequencing and quantitative validation. Please see lines 159-168.
- Line 240: Clearly define the criteria for identifying differentially expressed miRNAs (DEMs), including the fold-change cutoff and FDR threshold applied.
Responding: Thank you very much for your comment and suggestion. In this study, we utilized the DESeq2 analytical approach, which is grounded in the negative binomial distribution model, to perform comprehensive data analysis. The screening parameters for differential expression were set as p-value ≤ 0.05 and |log2(fold change)| ≥ 0.2. This whole paragraph has been revised, and please see lines 281-295.
- Line 327: Discuss whether the observed changes might suggest for a compensatory mechanism.
Responding: Thank you very much for your comment and suggestion. Flollowing your suggestion, we have conducted an initial and focused discussion here, while further in-depth discussions have been presented in the subsequent Discussion section. “The downregulation of SHMT appears to modulate the expression of specific miRNAs, potentially through its interaction with key enzymes within the miRNA processing path-way. Notably, the selective impact of SHMT downregulation on a limited subset of miRNAs suggests a targeted regulatory mechanism. Furthermore, while transcriptional alterations in genes encoding these miRNA-processing enzymes were observed following SHMT downregulation, the corresponding changes at the protein level remain to be elucidated. These findings underscore the need for further investigation into the precise molecular mechanisms underlying the selective influence of SHMT on miRNA expression, as well as the nature of the regulatory interplay between SHMT and these miRNAs.” Please see lines 409-418.
- Line 387: It is highly surprising that, miR-1174 and miR-1175 are upregulated by the depletion of SHMT both at 72h PIJ and 15h PBM. One of the corresponding author’s previous articles published in PNAS, 2014, showed that miR-1174 targets SHMT to maintain a low SHMT level in the mosquito gut by direct action on the SHMT transcript. It is not clear how both can be true. The authors should explain this in detail.
Responding: Thank you very much for your comments and suggestions. It has been demonstrated that miR-1174 targets SHMT, whereas miR-1175 does not. Located less than 100 bases apart on the genome, miR-1174 and miR-1175 form a cluster, originating from the same primary transcript and exhibiting similar expression patterns. In this study, we observed that both miR-1174 and miR-1175 were upregulated in response to SHMT downregulation, potentially due to the effect of SHMT downregulation on their common primary transcripts.
- Figure 1E: Replace RSP7 with RPS7.
Responding: Thank you very much for this suggestion. RSP7 has been replaced with RPS7.

Reviewer 2 Report
Comments and Suggestions for Authors
Review 2025- Serine hydroxymethyltransferase-mediated microRNAs in Aedes aegypti mosquitoes- Pu et al
Authors Pu et al investigate the mechanisms and impact of Serine hydroxymethyltransferase (SHMT), a crucial component of blood meal digestion impacting ovarian development, fecundity and flight capability in Aedes aegypti mosquitoes. While SHMT itself is expressed in low levels within the midgut, it was shown to impact digestive enzyme genes. This study evaluated several differentially expressed micro-RNAs in mosquitoes in response to SHMT downregulation and their potential role in suppressing female fecundity in Aedes. Further investigation of these mosquito mi-RNAs could provide a better understanding of biological processes for novel vector control. I have some minor comments from the authors-
- How many total mosquitoes were used in the study? (pool/triplicates?,n=?)
- Showing comparable data from SHMT-depleted and control mosquitoes at 72h PIJ established no significant impact on the process of injection. This was an excellent control shown and negates any questions regarding undue influence of the injection process.
- The study clearly documented contrasting morphological and quantitative differences between the experimental and control group 15 hr PBM. However, no reason is outlined for this particular time point. Eg. Was the morphology also checked at 13/17 hour points and then 15 h PBM selected for optimal visual differences?
- Fig. 2 legend does not describe what B and P stand (presumably before and post blood meal) for dsEGFP and dsSHMT. Surprisingly, while all the groups had a clear peak at 22 nucleotides for sRNA, the frequency % seemed to decline much sharply in control group (reaching as much as close to 50% in replicate 2-dsEGFP) compared to the experiment group. The authors should address this discrepancy.
- Interestingly, out of the four mi-RNAs further investigated for differential expression, only miR-2940-3p showed an impact on egg laying when downregulated. A direct role of this miRNA in fecundity, if established, would be of great interest to the biochemical and physiological process understanding for Ae. aegypti.
- Discrepancy in expression levels of Drosha and Dicer1 vs Dicer2 warrants further investigation. Understandably, as Dicer1 and Dicer2 are often associated with independent functions in organisms such as Drosophilla, this is not surprising.
Overall, this is a thoroughly conducted study on midgut physiology of Ae. aegypti outlining the crucial role played by SHMT. Further investigation of additional miRNAs involved in the process that may impact critical physiological processes in Aedes would provide researchers with a better understanding of the vector.
Author Response
- Authors Pu et al investigate the mechanisms and impact of Serine hydroxymethyltransferase (SHMT), a crucial component of blood meal digestion impacting ovarian development, fecundity and flight capability in Aedes aegypti mosquitoes. While SHMT itself is expressed in low levels within the midgut, it was shown to impact digestive enzyme genes. This study evaluated several differentially expressed micro-RNAs in mosquitoes in response to SHMT downregulation and their potential role in suppressing female fecundity in Aedes. Further investigation of these mosquito miRNAs could provide a better understanding of biological processes for novel vector control. I have some minor comments from the authors-
Responding: We extend our deepest gratitude for your invaluable time and meticulous review of our manuscript, despite your undoubtedly busy schedule. Your expert insights and constructive recommendations have been instrumental in guiding our revisions and enhancing the overall quality of our work. Following your esteemed advice, we have diligently revised the manuscript. Herein, we present our detailed, point-by-point responses to your comments.
- How many total mosquitoes were used in the study? (pool/triplicates?,n=?)
Responding: In this study, we conducted Small RNA sequencing at two distinct time points: before blood meal at 72 hours post-injection (72 h PIJ) and 15 hours post-blood meal (15 h PBM). For the small RNA sequencing, each treatment and control group included three biological replicates, with each replicate comprising the pooled total RNA from 6 mosquitoes.
- Showing comparable data from SHMT-depleted and control mosquitoes at 72h PIJ established no significant impact on the process of injection. This was an excellent control shown and negates any questions regarding undue influence of the injection process.
Responding: Thank you very much for your comments. Given the significant differences in the physiological activities of mosquitoes before and after blood feeding, selecting two time points—prior to blood feeding (72 hours post-injection, PIJ) and after blood feeding (15 hours post-blood meal, PBM)—for detection can more effectively demonstrate the molecular changes in mosquitoes induced by the downregulation of SHMT.
- The study clearly documented contrasting morphological and quantitative differences between the experimental and control group 15 hr PBM. However, no reason is outlined for this particular time point. Eg. Was the morphology also checked at 13/17 hour points and then 15 h PBM selected for optimal visual differences?
Responding: Thank you very much. In response to this valuable comment and suggestion, we have provided detailed explanations in the Methods section. Please see lines 159-168.
- Fig. 2 legend does not describe what B and P stand (presumably before and post blood meal) for dsEGFP and dsSHMT. Surprisingly, while all the groups had a clear peak at 22 nucleotides for sRNA, the frequency % seemed to decline much sharply in control group (reaching as much as close to 50% in replicate 2-dsEGFP) compared to the experiment group. The authors should address this discrepancy.
Responding: We appreciate your valuable comments and suggestions. B_dsEGFP and B_dsSHMT samples were collected 72 hours post-injection (PIJ), where B denotes pre-blood feeding. Meanwhile, P_dsEGFP and P_dsSHMT samples are being collected 15 hours post-blood meal (PBM), with P indicating post-blood feeding. This information has been incorporated into the figure legend. Please see lines 268-276.
The miRNA size distribution was primarily concentrated within the 21-22 nucleotide range (Bartel DP: Metazoan MicroRNAs. Cell 2018, 173(1):20-51.). Notably, the peak value of dsEGFP after 22 nt decreased by 50% in the control group; however, this reduction did not impact the overall miRNA peak value. The observed decrease may be attributed to other experimental factors such as sequencing or sampling procedures. Given that we conducted three biological replicates, stringent quality control measures were implemented during the sequencing process and data analysis, ensuring the authenticity and reliability of our data.
- Interestingly, out of the four miRNAs further investigated for differential expression, only miR-2940-3p showed an impact on egg laying when downregulated. A direct role of this miRNA in fecundity, if established, would be of great interest to the biochemical and physiological process understanding for Ae. aegypti.
Responding: Thank you for your valuable comments and suggestions. Currently, we are conducting systematic functional studies on several miRNAs identified in this study, with a particular focus on miR-2940-3p.
- Discrepancy in expression levels of Drosha and Dicer1 vs Dicer2 warrants further investigation. Understandably, as Dicer1 and Dicer2 are often associated with independent functions in organisms such as Drosophilla, this is not surprising.
Responding: We appreciate your valuable comments and suggestions. Currently, our findings indicate that the reduction in SHMT results in decreased levels of Drosha and Dicer1, while Dicer2 levels increase. However, the mechanisms underlying these changes require further investigation.
- Overall, this is a thoroughly conducted study on midgut physiology of Ae. aegypti outlining the crucial role played by SHMT. Further investigation of additional miRNAs involved in the process that may impact critical physiological processes in Aedes would provide researchers with a better understanding of the vector.
Responding: We appreciate your valuable comments and suggestions. Building upon our previous findings and current results, we are pursuing further investigations to elucidate the functional roles of miRNAs in critical physiological processes of mosquitoes, with particular emphasis on their potential involvement in midgut physiology.

Reviewer 3 Report
Comments and Suggestions for Authors
This well-written ms describes the characterization of Serine hydroxymethyltransferase in Aedes aegypti mosquitoes. SHMT is an important regulator of digestive enzymes in mosquitoes, is involved in glycine biosynthesis and thus has a reported association with a variety of other metabolic processes. The authors used miRNA profiling and qPCR of RNAi effectors to characterize the effects of SHMT silencing. This work is a follow-on from an earlier paper wherein mIR-1174 was found to control SHMT expression in mosquito guts.
The authors start the Introduction by explaining the biological significance of SHMT function. They then outline the importance of specific miRNAs in various metabolic processes. Their study relies on sRNA sequencing, qPCR and microscopy of ovarian phenotypes. It is not clear what the authors hoped to gain by characterizing the change in miRNA profiles upon SHMT silencing.
*It is difficult to effectively evaluate the reported results due to the absence of key details in the M&M section and figure legends.
Abstract-
Briefly summarize the biological importance of SHMT. Why study it?
Remove the italics from “Serine hydroxymethyltransferase”. Only gene names should be italicized.
What is ‘abnormal’ about the SHMT silencing phenotype?
Introduction
The gene accession numbers for protein coding and miRNA genes were not provided. Please do so.
Ln 28 “Serine hydroxymethyltransferase” is a protein name not a gene name, please remove the italics. Only gene symbols should be italicized.
Ln 38 Once again, what is ‘abnormal’ about the SHMT phenotype?
Ln 63-77 The overall hypothesis of this study is unclear.
Ln 67-68 Please clarify “The spatial discrepancy between SHMT and digestive enzyme genes implies that SHMT may indirectly modulate the biological function of the mosquito midgut.”
Materials and Methods
Ln 94-101 It appears that the authors chose to use raw PCR products for dsRNA synthesis rather than a cloned fragment of the SHMT gene. There is no indication that the sequence identity of the raw PCR products was validated prior to dsRNA production. This approach could lead to generation of dsRNAs of spurious amplifcation products rather than the GOI. A better approach would be to clone a ~500 bp fragment of SHMT coding sequence into a plasmid and use that for amplification of dsRNA.
Results
In general, the Results section is plagued by a lack of reporting of important details in figure legends, including the statistical test used and experimental details. In addition, across the board, different graphs in the same figure panel vary widely in the range of the y-axis, which make it very difficult to compare across treatment groups. Please standardize y-axes where possible and provide notes in the figure legend where this is not possible to alert the reader. Also, please indicate the number of replicates used to generate the figure.
The authors report that silencing of mIR-2940-3p results in reduced ovariole size and egg deposition, however they do not report whether differences occur in hatch rate. Moreover, this result had already been previously published, so it is unclear what new insight is provided here. What are the expected targets of mIR-2940-3p? A revised manuscript should contain this important information.
Ln 103-127 What method was used to calculate differential expression by qPCR? Please explain.
Results
Ln 197-206 This section should be moved to the methods section.
Fig 2 legend. What do 1, 2, 3 columns represent? Biological replicates? Please indicate this in the figure legend. What does B_eGFP represent? P_eGFP?
Ln 234-241 The authors state that select miRs “were notably downregulated in the dsSHMT group”. Please provide statistical details to support this assertion and all other similar assertions in this paragraph.
Fig 3 legend. Please define the acronyms used in the heatmaps. The sample order is flipped in Figure 3A vs 3B. Please make this consistent to ease reader viewing. Along these lines, Fig 3 legend needs to have details provided of DESeq2 FDR cut-offs. What units are represented in the scale bar?
Fig 4, Fig 5, Fig 7 legends. What calculation methods were used to determine miRNA differential expression? What do the different colored bars represent? What statistical test was used?
Ln 289 Please describe what you mean by ‘enhancement’ of the ovaries.
Fig 6F and 6I. How was blood feeding rate measured? Please add a section to the M&M section describing this.
What is Mic-2940-3p? How was it used? Please add a section to the M&M, including how these were made or manufacturers details.
Ln 314-335. Given that a purified plasmid was not used for SHMT dsRNA production, how can the authors be certain that the effects seen in Fig 7 are not due to non-specific dsRNAs present in the injection mix?
Ln 335- THe authors state that “These findings suggest that the effects of SHMT on miRNA biogenesis may be mediated through genes involved in the miRNA processing pathway.” There is no evidence presented in this ms that supports the idea that SHMT is directly involved with miRNA biogenesis, per se. The effects reported are more likely due to feedback and feed-forward reactions to SHMT depletion to overcome the physiological imbalance caused by gene silencing. In addition, due to the dsRNA preparation method, these results could also be due to off-target effects.
Ln 337 The authors state that SHMT is involved in pyrimidine biosynthesis. If that is the case, please provide another reference to back this up. The stated reference does not directly mention this association. Rather, the Jain paper, as cited, specifically ties SHMT to glycine metabolism.
Ln 338-339 The authors state that “A substantial body of 338 evidence has demonstrated that SHMT possesses diverse biological functions, including 339 roles in DNA synthesis, energy metabolism, and cell proliferation”. More accurately, SHMT has associations with these other processes. If SHMT actually has specific roles in the functions listed, please provide evidence.
Ln 361-363 Please clarify this statement , “This observation suggests that the aberrant biological function of the midgut in mosquitoes, resulting from down-regulation of SHMT, may be indirectly 362 mediated through its influence on other genes or molecular pathways.” It goes without saying that physiological processes are upset by SHMT silencing. Please be more precise. What is mediated?
Ln 363-368 Please move this section to the introduction.
Ln 369-380 Since silencing of the 3 of the 4 ‘major’ DEMs did not result in detrimental effects of bloodmeal digestion or associated processes like egg-laying, what is your hypothesis as to why they were modulated upon SHMT silencing?
Ln 387-401 The authors provide a paragraph summarizing biological relevance of mIRs used in this study, however these mIRs were not found to have significant effects in the context of this work. So, this information seems off-topic. Instead, I recommend identification of possible targets of mIR-2940-3p and discussion of the implications of those results.
Ln 416-418 The authors speculate that SHMT is responsible for miRNA biogenesis. If that were true, then one would expect that all mIRs assessed in this work would be downregulated, and that is not the case. Therefore, they need to rephrase their speculation about the reduced levels of miRNA biogenesis coding gene transcripts upon SHMT silencing.
Author Response
- This well-written ms describes the characterization of Serine hydroxymethyltransferase in Aedes aegyptimosquitoes. SHMT is an important regulator of digestive enzymes in mosquitoes, is involved in glycine biosynthesis and thus has a reported association with a variety of other metabolic processes. The authors used miRNA profiling and qPCR of RNAi effectors to characterize the effects of SHMT silencing. This work is a follow-on from an earlier paper wherein mIR-1174 was found to control SHMT expression in mosquito guts.
Responding: We sincerely appreciate your time and effort in reviewing our manuscript and providing invaluable suggestions for improvement. We have carefully revised the manuscript in accordance with your insightful comments. Your expert recommendations have been instrumental in significantly enhancing the quality of our work.
- The authors start the Introduction by explaining the biological significance of SHMT function. They then outline the importance of specific miRNAs in various metabolic processes. Their study relies on sRNA sequencing, qPCR and microscopy of ovarian phenotypes. It is not clear what the authors hoped to gain by characterizing the change in miRNA profiles upon SHMT silencing.
Responding: Thank you very much for your comments and suggestions. We fully agree with your opinion. We have revised the Introduction, and the modified version is more coherent in expression, clearer in logic, and more distinct in theme. The goal of this study is to identify SHMT-responsive miRNAs through small RNA sequencing, aiming to provide new insights for further research into the function and mechanisms of SHMT in mosquitoes. Please see lines 31-96.
- It is difficult to effectively evaluate the reported results due to the absence of key details in the M&M section and figure legends.
Responding: Thank you very much for your comments and suggestions. Your concern is well-founded, and we have already made revisions to the methodology section and figure legends.
- Abstract- Briefly summarize the biological importance of SHMT. Why study it?
Responding: Thank you very much for your comments and suggestions. We have added the importance of SHMT in the summary. Please see lines 11-12.
- Remove the italics from “Serine hydroxymethyltransferase”. Only gene names should be italicized.
Responding: Thank you very much for your suggestion. We have removed the italics from “Serine hydroxymethyltransferase”.
- What is ‘abnormal’ about the SHMT silencing phenotype?
Responding: Thank you very much for your question. The “abnormal phenotype” has been changed to “midgut physiology”. The abnormal phenotype here refers to the obstruction of blood digestion in the gut of mosquitoes. The statement has been revised in Abstract.
- Introduction-The gene accession numbers for protein coding and miRNA genes were not provided. Please do so.
Responding: Thank you very much for your question. The gene accession numbers have been provided.
- Ln 28 “Serine hydroxymethyltransferase” is a protein name not a gene name, please remove the italics. Only gene symbols should be italicized.
Responding: Thank you very much for your question. The italics has been removed from the “Serine hydroxymethyltransferase”.
- Ln 38 Once again, what is ‘abnormal’ about the SHMT phenotype?
Responding: Thank you very much for your question. Also, the “abnormal phenotype” has been changed to “midgut physiology”. The abnormal phenotype here also refers to the obstruction of blood digestion in the mosquito gut.
- Ln 63-77 The overall hypothesis of this study is unclear.
Responding: Thank you very much. In response, we have thoroughly revised this section. Please check the final two paragraphs of the introduction for the updated content. Details are in lines 78-96.
- Ln 67-68 Please clarify “The spatial discrepancy between SHMT and digestive enzyme genes implies that SHMT may indirectly modulate the biological function of the mosquito midgut.”
Responding: We sincerely appreciate your valuable suggestion. SHMT is expressed at low levels in the midgut but at significantly higher levels in other tissues, particularly in the fat body. In contrast, digestive enzymes are exclusively expressed in the midgut and not detected in other tissues. This spatial segregation of SHMT and digestive enzyme expression suggests that SHMT may indirectly influence the biological functions of the midgut in mosquitoes. The statement has been revised. Please see lines 78-80.
- Materials and Methods-Ln 94-101 It appears that the authors chose to use raw PCR products for dsRNA synthesis rather than a cloned fragment of the SHMT gene. There is no indication that the sequence identity of the raw PCR products was validated prior to dsRNA production. This approach could lead to generation of dsRNAs of spurious amplifcation products rather than the GOI. A better approach would be to clone a ~500 bp fragment of SHMT coding sequence into a plasmid and use that for amplification of dsRNA.
Responding: We sincerely appreciate your valuable comments and suggestions. In our synthesis of double-stranded RNA, PCR products were utilized as templates. These templates were specifically amplified using primers designed for SHMT, and the resulting PCR products were verified by sequencing to ensure they correspond exclusively to SHMT fragment sequence. Therefore, we think that the dsRNA (dsSHMT) used for injection should be specific to SHMT, eliminating the possibility of incorporating dsRNA from other genes. We also added corresponding information in the method section. Please see lines 124-125.
13-Results -In general, the Results section is plagued by a lack of reporting of important details in figure legends, including the statistical test used and experimental details. In addition, across the board, different graphs in the same figure panel vary widely in the range of the y-axis, which make it very difficult to compare across treatment groups. Please standardize y-axes where possible and provide notes in the figure legend where this is not possible to alert the reader. Also, please indicate the number of replicates used to generate the figure.
Responding: We sincerely appreciate your valuable comments and suggestions. The details you highlighted are of critical importance. Following your recommendations, we have thoroughly revised all figures and figure legends accordingly.
- The authors report that silencing of mIR-2940-3p results in reduced ovariole size and egg deposition, however they do not report whether differences occur in hatch rate. Moreover, this result had already been previously published, so it is unclear what new insight is provided here. What are the expected targets of mIR-2940-3p? A revised manuscript should contain this important information.
Responding: We sincerely appreciate your valuable comments and suggestions. Your comments and suggestions are undoubtedly helpful for us to improve the quality of our manuscript. We fully agree with your opinions. We are aware that the functions of SHMT and miR-2940-3p have been previously reported, and the phenotypes resulting from changes in their expression are not the innovative points of this paper. The goal of this paper is not to reveal the function of a specific miRNA, but to identify the SHMT-responsive miRNAs. Based on the identification of these miRNAs, we have preliminarily studied the functions of several miRNAs, among which the phenotype observed after altering miR-2940-3p is largely consistent with previous reports. Although the function of miR-2940-3p has been reported, we believe that the functions and mechanisms of these miRNAs still require further in-depth research. Therefore, building on this study, we are continuing to investigate the biological functions and mechanisms of these miRNAs in the life activities of mosquitoes. However, due to time constraints, we anticipate that these results will be published in our subsequent articles.
- Ln 103-127 What method was used to calculate differential expression by qPCR? Please explain.
Responding: We sincerely appreciate your suggestion. To prevent redundancy, we have included the calculation method in M&M 2.7: “All statistical analyses were conducted using GraphPad Prism 8.0 (GraphPad Soft-ware, La Jolla, CA, USA), with data presented as mean ± standard error of the mean (SEM). Each experiment was independently replicated a minimum of three times, with each rep-lication comprising at least three technical replicates. Error bars represent variability among experimental replicates. The mean cycle threshold (Ct) values of RT-qPCR assay were transformed into relative expression levels through application of the 2-ΔΔCt method [41-43]. Comparative analysis of expression levels was performed using a two-tailed un-paired Student's t-test. Statistical significance thresholds were established as follows: *p < 0.05, **p < 0.01, ***p < 0.001, ****p < 0.0001, and ns indicates “not significant” (p > 0.05).” Please see lines 156-157 and 217-226.
- Results-Ln 197-206 This section should be moved to the methods section.
Responding: Thank you very much for your suggestion. Following your suggestion, this section has been removed and combined with the methods section 2.5.1. Quality control of sequencing data. Please see lines 174-185.
- Fig 2 legend. What do 1, 2, 3 columns represent? Biological replicates? Please indicate this in the figure legend. What does B_eGFP represent? P_eGFP?
Responding: Thank you very much for your suggestion. This information has been added to the figure legend. “There were three biological replicates in both the experimental group (dsSHMT) and the control group (dsEGFP), with each replicate consisting of total RNA or small RNA extracted from eight mosquitoes. The numbers 1, 2, and 3 in the columns represent three biological replicates. In the sample labels, "B" indicates samples collected before blood meal at 72 hours post injection (72 h PIJ), while "P" indicates samples collected at 15 hours after blood feeding (15 h PBM). B_dsEGFP and B_dsSHMT represent the samples collected before blood feeding, with B_dsSHMT being the treated group and B_dsEGFP being the control group. P_dsEGFP and P_dsSHMT represent the samples collected after blood feeding, with P_dsSHMT being the treated group and P_dsEGFP being the control group.” Please see lines 270-278.
- Ln 234-241 The authors state that select miRs “were notably downregulated in the dsSHMT group”. Please provide statistical details to support this assertion and all other similar assertions in this paragraph.
Responding: Thank you very much for your suggestion. We have added two tables into the manuscript, which clearly present the differential expression screening results of miRNAs at the two time points. The screening criteria for differentially expressed miRNAs were set as follows: DESeq2 p-value ≤ 0.05 and |log2(fold change)| ≥ 0.2. The information has been updated in this paragraph. Please see lines 283-298.
- Fig 3 legend. Please define the acronyms used in the heatmaps. The sample order is flipped in Figure 3A vs 3B. Please make this consistent to ease reader viewing. Along these lines, Fig 3 legend needs to have details provided of DESeq2 FDR cut-offs. What units are represented in the scale bar?
Responding: Thank you very much for your suggestion. The acronyms used in the heatmaps have been defined in the figure legend. The sample order in this figure has been adjusted. The cutoffs have been added to the figure legend. Red indicates high expression, while blue indicates low expression. The bar gradient from red to blue represents the range of log10(TPM+1) values from high to low. Please see lines 302-312.
- Fig 4, Fig 5, Fig 7 legends. What calculation methods were used to determine miRNA differential expression? What do the different colored bars represent? What statistical test was used?
Responding: Thank you very much for these questions. The calculation methods were included in the M&M section 2.7, “All statistical analyses were conducted using GraphPad Prism 8.0 (GraphPad Soft-ware, La Jolla, CA, USA), with data presented as mean ± standard error of the mean (SEM). Each experiment was independently replicated a minimum of three times, with each rep-lication comprising at least three technical replicates. Error bars represent variability among experimental replicates. The mean cycle threshold (Ct) values of RT-qPCR assay were transformed into relative expression levels through application of the 2-ΔΔCt method [41-43]. Comparative analysis of expression levels was performed using a two-tailed un-paired Student's t-test. Statistical significance thresholds were established as follows: *p < 0.05, **p < 0.01, ***p < 0.001, ****p < 0.0001, and ns indicates “not significant” (p > 0.05).” Following your recommendations, we have also incorporated this information into the figure legends.
Since the horizontal axis is clearly labeled, the colors serve more as a visual aid rather than conveying critical information. However, for clarity and consistency, we can interpret the colors as follows: orange represents the wild type (WT), purple represents the dsEGFP control group, pink represents the up-regulated dsSHMT group, green represents the down-regulated dsSHMT group, and brown represents the group with unchanged dsSHMT expression levels.
- Ln 289 Please describe what you mean by ‘enhancement’ of the ovaries.
Responding: Thank you very much for your question. Our previous explanation lacked clarity and precision in wording. The previous statement has been changed to “However, compared to the control groups, the injection of miR-2940-3p mimic resulted in slightly larger size observed at the same time point.” Please see lines 361-362.
- Fig 6F and 6I. How was blood feeding rate measured? Please add a section to the M&M section describing this.
Responding: Thank you very much for your question and suggestion. The calculation method for the blood-feeding rate in Fig 6F and 6I is as follows: To assess the blood-feeding rate, three independent cages were established for both control and treatment groups, representing three biological replicates. Each cage contained a population of 40 female mosquitoes accompanied by a minimum of 20 male mosquitoes. Following the blood-feeding period, the number of engorged females in each cage was recorded. The blood-feeding rate for each replicate was calculated using the following equation: Blood-feeding rate (%) = (Number of blood-fed females / Total number of females) × 100. Statistical analysis was subsequently performed using GraphPad Prism 8 (GraphPad Software, San Diego, CA, USA). These statements have been added to the M&M section. Please see lines 107-115.
- What is Mic-2940-3p? How was it used? Please add a section to the M&M, including how these were made or manufacturers details.
Responding: Thank you very much for your suggestion. Mic-2940-3p is a chemically synthesized mimic of miR-2940-3p, presented as a double-stranded molecule. miRNA mimics can replicate the functions of endogenous miRNAs, enabling transient upregulation of miRNA activity. This information has been included in the Methods section. Please see lines .211-217.
- Ln 314-335. Given that a purified plasmid was not used for SHMT dsRNA production, how can the authors be certain that the effects seen in Fig 7 are not due to non-specific dsRNAs present in the injection mix?
Responding: Thank you very much for your comment. When synthesizing the double-stranded RNA, we first synthesize the cDNA, and then use SHMT-specific primers for PCR to synthesize the DNA. The PCR product was gel-purified and sequenced, confirming it to be an SHMT-specific template. Therefore, transcription using this DNA template can only produce double-stranded RNA specific to SHMT. This information has been included in the Methods section. Please see lines 123-125.
- Ln 335- The authors state that “These findings suggest that the effects of SHMT on miRNA biogenesis may be mediated through genes involved in the miRNA processing pathway.” There is no evidence presented in this ms that supports the idea that SHMT is directly involved with miRNA biogenesis, per se. The effects reported are more likely due to feedback and feed-forward reactions to SHMT depletion to overcome the physiological imbalance caused by gene silencing. In addition, due to the dsRNA preparation method, these results could also be due to off-target effects.
Responding: We appreciate your valuable comments and suggestions. Following your suggestion, we have revised the Discussion section. In this study, we did not establish how SHMT influences the expression of these enzymes or elucidate the specific mechanisms by which SHMT affects the expression of these miRNAs. Currently, our understanding is limited to speculative hypotheses based on observed changes in gene expression. Clearly, further investigation and more robust evidence are required to clarify the molecular mechanisms underlying the downregulation of SHMT and its impact on gene expression.
- Ln 337 The authors state that SHMT is involved in pyrimidine biosynthesis. If that is the case, please provide another reference to back this up. The stated reference does not directly mention this association. Rather, the Jain paper, as cited, specifically ties SHMT to glycine metabolism.
Responding: We appreciate your valuable comments and suggestions. To ensure the discussion remains sharply focused on the topic of this study, we have removed some contents in the first paragraph of the previous manuscript.
- Ln 338-339 The authors state that “A substantial body of evidence has demonstrated that SHMT possesses diverse biological functions, including roles in DNA synthesis, energy metabolism, and cell proliferation”. More accurately, SHMT has associations with these other processes. If SHMT actually has specific roles in the functions listed, please provide evidence.
Responding: We appreciate your valuable comments and suggestions. Numerous studies have demonstrated the functions of SHMT, and some evidences are shown below. However, to maintain a focused discussion on the topic of this study, we have deleted some contents in the first paragraph of the previous manuscript.
Extensive research has established that SHMT is involved in multiple biological processes, including DNA synthesis, energy metabolism, and cell proliferation [1-7]. (1) As a pivotal enzyme in carbon metabolism, SHMT plays a crucial role in DNA synthesis. During the S phase of the cell cycle, cells require substantial amounts of deoxythymidine monophosphate (dTMP) to facilitate DNA replication. The nuclear transport of three key enzymes - thymidylate synthase (TYMS), dihydrofolate reductase (DHFR), and SHMT1 - is mediated through small ubiquitin-like modifier (SUMO) modification. These SUMOylated enzymes subsequently assemble to form the thymidylate synthesis complex (dTMP-SC) within the nucleus, enabling efficient dTMP production [8-10]. In the cytoplasm, 5,10-methylene-tetrahydrofolate (5,10-CH2-THF), catalyzed by SHMT1, functions as a cofactor for thymidylate synthase (TYMS). It catalyzes the methylation of deoxyuridine monophosphate (dUMP) to produce deoxythymidylate (dTMP) and dihydrofolate (DHF). Subsequently, dihydrofolate reductase (DHFR) reduces DHF to tetrahydrofolate (THF), maintaining the folate cycle, which is a critical link in DNA synthesis and repair [11-14]. (2) In the regulation of energy metabolism, SHMT2 serves as a critical mitochondrial enzyme that orchestrates key metabolic processes. Experimental evidence indicates that SHMT2 deficiency disrupts mitochondrial bioenergetics, significantly impairing oxidative phosphorylation capacity and subsequently attenuating cellular proliferation [15]. Furthermore, through its integral involvement in one-carbon metabolic pathways, SHMT2 exerts regulatory control over NADPH biosynthesis, thereby playing an indispensable role in sustaining cellular energy metabolism and preserving redox balance [16]. (3) Regarding cell proliferation, SHMT2 exhibits high expression levels across various types of tumors and cancers, demonstrating a significant association with enhanced cellular proliferation, migration, and invasion. For instance, in colorectal cancer (CRC) cells, SHMT2, as a target gene of β-catenin, facilitates tumor cell proliferation and metastasis by inhibiting the ubiquitination and degradation of β-catenin [17]. Additionally, in hepatocellular carcinoma (HCC), SHMT1 suppresses HCC proliferation and metastasis through inhibition of reactive oxygen species (ROS) production mediated by NADPH oxidase 1 (NOX1) [18]. In summary, SHMT proteins play multifaceted and crucial roles in cellular physiology, particularly in DNA synthesis, energy metabolism regulation, and the control of cell proliferation. Their diverse functions and regulatory mechanisms highlight their significance in both normal cellular processes and pathological conditions.
Reference for the evidence:
- McBride MJ, Hunter CJ, Zhang Z, TeSlaa T, Xu X, Ducker GS, Rabinowitz JD: Glycine homeostasis requires reverse SHMT flux. Cell Metab 2024, 36(1):103-115 e104.
- Ron-Harel N, Santos D, Ghergurovich JM, Sage PT, Reddy A, Lovitch SB, Dephoure N, Satterstrom FK, Sheffer M, Spinelli JB et al: Mitochondrial Biogenesis and Proteome Remodeling Promote One-Carbon Metabolism for T Cell Activation. Cell Metab 2016, 24(1):104-117.
- Renwick SB, Snell K, Baumann U: The crystal structure of human cytosolic serine hydroxymethyltransferase: a target for cancer chemotherapy. Structure 1998, 6(9):1105-1116.
- Jin Y, Jung SN, Lim MA, Oh C, Piao Y, Kim HJ, Nguyena Q, Kang YE, Chang JW, Won HR et al: SHMT2 Induces Stemness and Progression of Head and Neck Cancer. Int J Mol Sci 2022, 23(17).
- Wang W, Wang M, Du T, Hou Z, You S, Zhang S, Ji M, Xue N, Chen X: SHMT2 Promotes Gastric Cancer Development through Regulation of HIF1alpha/VEGF/STAT3 Signaling. Int J Mol Sci 2023, 24(8).
- Sun M, Zhao M, Li R, Zhang Y, Shi X, Ding C, Ma C, Lu J, Yue X: SHMT2 promotes papillary thyroid cancer metastasis through epigenetic activation of AKT signaling. Cell Death Dis 2024, 15(1):87.
- Zhang S, He R, Zhang M, Zhang J, Wu M, Zhang G, Jiang T: Elucidation of the Role of SHMT2 in L-Serine Homeostasis in Hypoxic Hepa1-6 Cells. Int J Mol Sci 2024, 25(21).
- Anderson DD, Woeller CF, Stover PJ: Small ubiquitin-like modifier-1 (SUMO-1) modification of thymidylate synthase and dihydrofolate reductase. Clin Chem Lab Med 2007, 45(12):1760-1763.
- Kamynina E, Lachenauer ER, DiRisio AC, Liebenthal RP, Field MS, Stover PJ: Arsenic trioxide targets MTHFD1 and SUMO-dependent nuclear de novo thymidylate biosynthesis. Proc Natl Acad Sci U S A 2017, 114(12):E2319-E2326.
- Woeller CF, Anderson DD, Szebenyi DM, Stover PJ: Evidence for small ubiquitin-like modifier-dependent nuclear import of the thymidylate biosynthesis pathway. J Biol Chem 2007, 282(24):17623-17631.
- Anderson DD, Stover PJ: SHMT1 and SHMT2 are functionally redundant in nuclear de novo thymidylate biosynthesis. PLoS One 2009, 4(6):e5839.
- Spizzichino S, Boi D, Boumis G, Lucchi R, Liberati FR, Capelli D, Montanari R, Pochetti G, Piacentini R, Parisi G et al: Cytosolic localization and in vitro assembly of human de novo thymidylate synthesis complex. FEBS J 2022, 289(6):1625-1649.
- Anderson DD, Eom JY, Stover PJ: Competition between Sumoylation and Ubiquitination of Serine Hydroxymethyltransferase 1 Determines Its Nuclear Localization and Its Accumulation in the Nucleus. J Biol Chem 2012, 287(7):4790-4799.
- Lee SW, Chen TJ, Lin LC, Li CF, Chen LT, Hsing CH, Hsu HP, Tsai CJ, Huang HY, Shiue YL: Overexpression of thymidylate synthetase confers an independent prognostic indicator in nasopharyngeal carcinoma. Exp Mol Pathol 2013, 95(1):83-90.
- Fiddler JL, Blum JE, Heyden KE, Castillo LF, Thalacker-Mercer AE, Field MS: Impairments in SHMT2 expression or cellular folate availability reduce oxidative phosphorylation and pyruvate kinase activity. Genes Nutr 2023, 18(1):5.
- Jie H, Wei J, Li Z, Yi M, Qian X, Li Y, Liu C, Li C, Wang L, Deng P et al: Serine starvation suppresses the progression of esophageal cancer by regulating the synthesis of purine nucleotides and NADPH. Cancer Metab 2025, 13(1):10.
- Liu C, Wang L, Liu X, Tan Y, Tao L, Xiao Y, Deng P, Wang H, Deng Q, Lin Y et al: Cytoplasmic SHMT2 drives the progression and metastasis of colorectal cancer by inhibiting beta-catenin degradation. Theranostics 2021, 11(6):2966-2986.
- Dou C, Xu Q, Liu J, Wang Y, Zhou Z, Yao W, Jiang K, Cheng J, Zhang C, Tu K: SHMT1 inhibits the metastasis of HCC by repressing NOX1-mediated ROS production. J Exp Clin Cancer Res 2019, 38(1):70.
- Ln 361-363 Please clarify this statement, “This observation suggests that the aberrant biological function of the midgut in mosquitoes, resulting from down-regulation of SHMT, may be indirectly mediated through its influence on other genes or molecular pathways.” It goes without saying that physiological processes are upset by SHMT silencing. Please be more precise. What is mediated?
Responding: We sincerely appreciate your valuable suggestion and have accordingly revised the relevant section. Please see lines 444-448.
- Ln 363-368 Please move this section to the introduction.
Responding: We are grateful for your insightful suggestion. In response, we have relocated this section to the Introduction and made corresponding revisions to ensure coherence throughout the introductory content. Please see lines 51-69.
- Ln 369-380 Since silencing of the 3 of the 4 ‘major’ DEMs did not result in detrimental effects of bloodmeal digestion or associated processes like egg-laying, what is your hypothesis as to why they were modulated upon SHMT silencing?
Responding: Thank you very much for this question. When the expression levels of these miRNAs were experimentally modulated, no significant phenotypic alterations were observed in mosquitoes. This phenotypic resilience could potentially be attributed to multiple underlying factors: (1) the magnitude of miRNA expression alteration might have been insufficient to elicit detectable phenotypic changes; (2) these miRNAs may exert subtle regulatory functions that are not readily apparent at the organismal level; and (3) potential activation of compensatory molecular pathways within the organism may have mitigated the effects of miRNA modulation. Considering the highly specific and complex nature of miRNA functions, it is crucial to employ advanced investigative approaches, particularly gene editing technologies, to further elucidate their biological roles. Quantitative PCR analysis revealed that SHMT downregulation significantly influenced the expression profiles of specific miRNAs, indicating a potential indirect and selective regulatory role of SHMT in miRNA biogenesis or expression modulation. Nevertheless, the precise molecular mechanisms underlying this regulatory relationship remain to be fully characterized and warrant further comprehensive investigation.
Following your suggestion, we have revised the discussion section accordingly. Please see lines 464-469.
- Ln 387-401 The authors provide a paragraph summarizing biological relevance of mIRs used in this study, however these mIRs were not found to have significant effects in the context of this work. So, this information seems off-topic. Instead, I recommend identification of possible targets of mIR-2940-3p and discussion of the implications of those results.
Responding: We sincerely appreciate your valuable comments and suggestions. We acknowledge that the mentioned paragraph is not pertinent to our research topic. After thorough deliberation, we have decided to remove this section. Regarding your suggestion to identify the target genes of miR-2940-3p and discuss their functions, we fully concur with your recommendation. However, due to time limitations, we plan to explore and report on the regulatory role of miR-2940-3p and its additional target genes in future publications.
- Ln 416-418 The authors speculate that SHMT is responsible for miRNA biogenesis. If that were true, then one would expect that all mIRs assessed in this work would be downregulated, and that is not the case. Therefore, they need to rephrase their speculation about the reduced levels of miRNA biogenesis coding gene transcripts upon SHMT silencing.
Responding: We sincerely appreciate your valuable comments and suggestions, which have guided our revisions. Please see lines 489-498.
Down-regulation of SHMT led to a decrease in the expression levels of Drosha, Dicer1, and AGO1, while inducing up-regulation of Dicer2 and AGO2 (Figure 7). These findings suggest that SHMT potentially modulates miRNA expression by influencing key enzymes involved in the miRNA biogenesis pathway. However, among the miRNAs identified in this study, only a subset exhibited significant expression changes, including both up-regulated and down-regulated species. This selective pattern of miRNA regulation indicates that SHMT may specifically target certain miRNAs rather than exerting a global effect on miRNA expression. The production and expression of these SHMT-responsive miRNAs are likely influenced by additional regulatory factors, suggesting a complex regulatory network. Therefore, the precise molecular mechanisms underlying SHMT-mediated regulation of specific miRNAs warrant further comprehensive investigation.

Round 2
Reviewer 3 Report
Comments and Suggestions for Authors
The authors have responded to most reviewer comments.
Here are are few minor items.
Ln 10- Mosquitoes transmit pathogens not ‘diseases’.
Ln 33- Please change to “These pathogens are responsible for…..” instead of “These diseases…”
Fig 2 legend. Please change the Fig legend to say, “Each column represents an individual biological replicate”.
Author Response
The authors have responded to most reviewer comments.
Here are are few minor items.
Ln 10- Mosquitoes transmit pathogens not ‘diseases’.
Responding: Changed. Please see line 10.
Ln 33- Please change to “These pathogens are responsible for…..” instead of “These diseases…”
Responding: Changed. Please see line 33.
Fig 2 legend. Please change the Fig legend to say, “Each column represents an individual biological replicate”.
Responding: Changed. Please see lines 274-275.